# The evolutionary dynamics between viral mimics and host proteins

Rotem Fuchs [1,5], Ofir Schor [1,5], Bar Naim[1,5], Dafna Tussia-Cohen [1], Alessandra Mozzi [2], Diego Forni[2], Sivan Friedman [1], Zohar Haggai [1], Manuela Sironi [3,4] & Tzachi Hagai [1]✉

## Abstract

**Viral proteins interact with host proteins to hijack cellular pathways important for viral replication. Viral mimics are proteins whose structural similarity to host-mimicked proteins allows them to interact with mutual host targets. This mimicry poses a challenge for the host—how to avoid mimics without compromising essential interactions with host-mimicked proteins. Despite the prevalence of mimicry, the evolutionary dynamics between host and viral mimics remain largely unknown. We address this by integrating structural modeling, host–virus interaction networks, and comprehensive evolutionary analyses of host and viral proteins. We show that host proteins targeted by mimics and host-mimicked proteins are highly conserved, and that this is related to functional constraints imposed on host proteins. Host interface residues that interact with both mimics and host-mimicked proteins evolve slowly, while residues that exclusively interact with mimics evolve significantly faster. Surprisingly, viral mimics do not evolve rapidly, instead displaying complex evolutionary patterns. Our analysis reveals host's limited capacity to escape mimicry and viral evolution to exploit this, and highlights how constraints lead to unexpectedly slow evolution of host–virus interaction networks.**

**Keywords** Molecular Mimicry; Host-Virus Interactions; Protein-Protein Interactions; Virus Evolution; Structural modeling
**Subject Categories** Evolution & Ecology; Microbiology, Virology & Host Pathogen Interaction

## Introduction

For successful replication, viral proteins must form numerous interactions with host proteins to hijack and rewire cellular pathways. During their evolution, viruses acquired genes from their hosts through horizontal gene transfer, which were then repurposed for various functions, including host modulation (Koonin et al, 2022). An important class of these acquisitions involves protein domains that retain an overall structural similarity between the original host (hereafter, host-mimicked protein) and the acquired viral domain (hereafter, viral mimic) (Elde and Malik, 2009). Domain mimicry (that differs from epitope mimicry that occurs at the primary sequence level (Maguire et al, 2024)) is a prevalent mechanism utilized by many viruses for co-option or disruption of host machineries through direct host–virus protein-protein interactions (PPIs) (Elde and Malik, 2009; Lasso et al, 2021; Boys et al, 2023). This utility stems from the similarity in the interaction interface between the viral-mimicking and host-mimicked domains, allowing the viral mimic to bind the same set of proteins as the host's. Host proteins that both the host-mimicked and the viral-mimicking proteins interact with, through interface similarity, are termed mutual host targets (see Fig. 1A for a schematic representation). For example, at least five unrelated viral families encode Bcl-2 domains that mimic host Bcl-2s to interact with specific host targets and inhibit apoptosis (Kvansakul and Hinds, 2013; Kvansakul et al, 2017).

Host–virus PPIs pose a strong selection pressure on both the host and the virus, since they confer a selective advantage for one of the interacting partners while being deleterious for the other (Duggal and Emerman, 2012; Sironi et al, 2015; Enard et al, 2016; Tenthorey et al, 2022). These antagonistic interactions can lead to an evolutionary arms race, where both sides evolve to retain or escape these interactions by substitutions of residues at the interface between host and viral proteins. This evolutionary dynamics is observed in many cases by detecting signatures of positive selection in residues related to these interactions in both host- and viral-interacting proteins (Duggal and Emerman, 2012; Sawyer and Elde, 2012; Demogines et al, 2013; Tsu et al, 2021). However, in evolutionary conflicts involving mimicry, hosts are limited in their evolutionary capacity to escape interactions with mimics since evolving away from these interactions can compromise the host-mimicked protein's cellular interaction network and reduce host fitness (Elde and Malik, 2009; Shuler and Hagai, 2022). Furthermore, the viral mimic may also be constrained by the requirement to retain similarity in structure and function to the host-mimicked protein. This is further complicated by the existence of mimics that diverge in function from the original host function (Mutz et al, 2023; Graham et al, 2008; Dower, 2000) and by duplication of some of these mimics in the viral genome(Senkevich

[1]Shmunis School of Biomedicine and Cancer Research, George S Wise Faculty of Life Sciences, Tel Aviv University, Tel Aviv 69978, Israel. [2]Scientific Institute IRCCS E. MEDEA, Computational Biology Unit, Bosisio Parini 23842, Italy. [3]School of Medicine and Surgery, University of Milano-Bicocca, Monza (MB) 20900, Italy. [4]Scientific Institute IRCCS San Gerardo dei Tintori Foundation, Monza (MB) 20900, Italy. [5]These authors contributed equally: Rotem Fuchs, Ofir Schor, Bar Naim. ✉E-mail: tzachiha@tauex.tau.ac.il

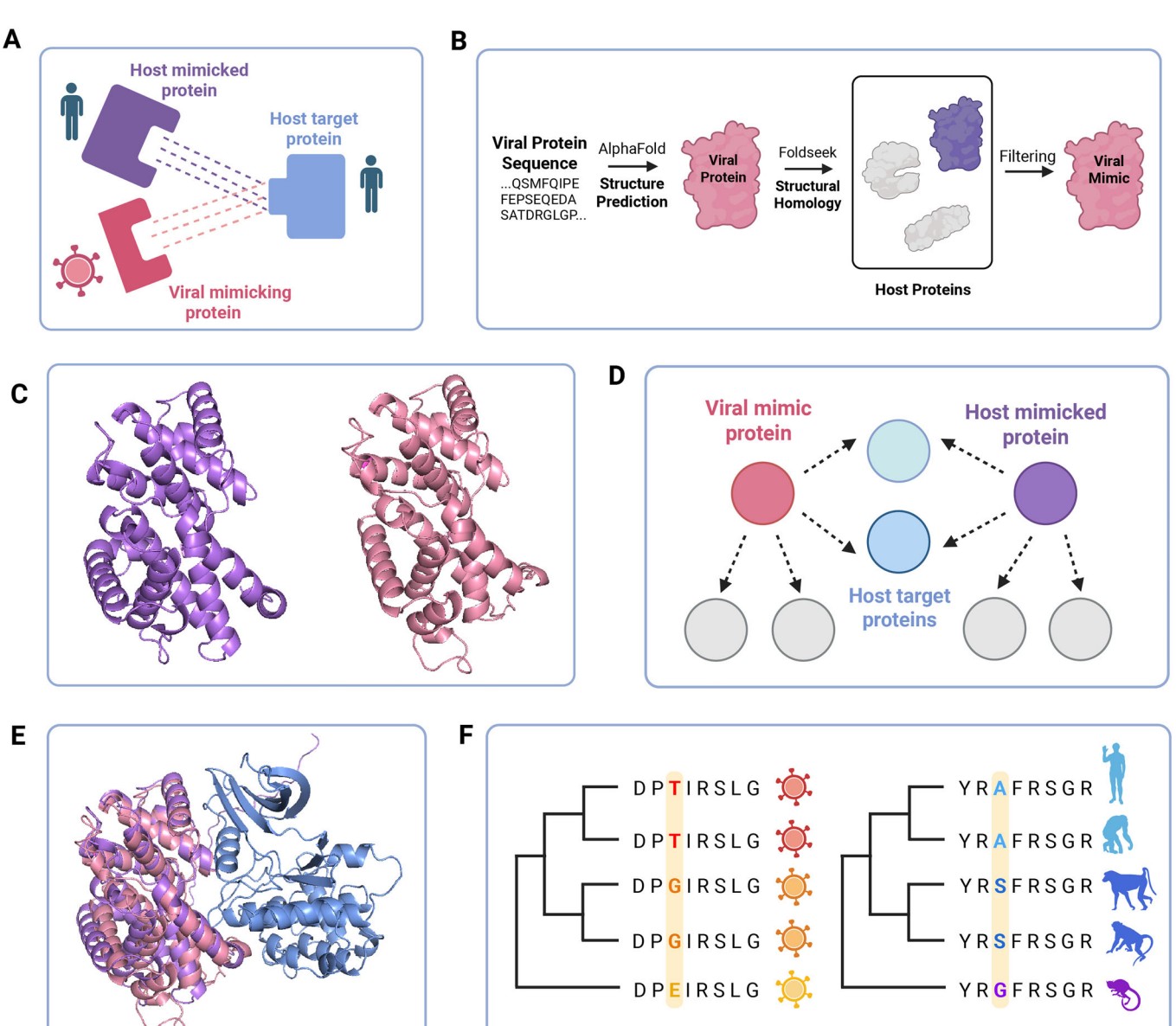

**Figure 1. System description and study design.**

(A) An illustration of the three proteins involved in domain mimicry interactions. Viral mimics (in pink) have domains that structurally resemble a domain found in a host protein (in purple). Both host-mimicked and viral-mimicking proteins interact with a mutual host target (in blue) using a similar binding region. (B) Computational workflow for detecting viral domain mimics. The structures of proteins from five human-infecting DNA viruses were predicted using AlphaFold. Structural homolog pairs of human-mimicked and viral-mimicking proteins were identified using Foldseek and additional filtering to avoid spurious matching. (C) An example of a detected structural homology between viral ORF72 (from KSHV, in pink) and human cyclin CCNO (purple). (D) Human proteins that interact with both viral-mimicking and human-mimicked proteins were identified using human–virus and within-human PPI maps. These human proteins are referred to as mutual host targets. (E) An example of a viral-mimicking, host-mimicked and mutual target triad, with ORF72 (pink), CCNO (purple) and CDK3 (blue), respectively. Protein complex structures of (1) host-mimicked– target and (2) viral-mimicking–target, were predicted by AlphaFold-Multimer and superimposed based on the target. The disordered 80AA N-terminus tail of CCNO is removed for visualization. (F) Several phylogeny-based approaches and models were used to infer host and viral gene evolutionary rates, relative rates of residues within each protein, and to detect signatures of positive selection. Images were created in BioRender.(2025) https://BioRender.com/1m65sw7. Silhouette images were derived from https://www.phylopic.org/./pp/p. Structure images were created using PyMol (https://www.pymol.org/). Source data are available online for this figure.

et al, 2021; Bratke et al, 2013), suggesting complex evolutionary patterns of viral-encoded mimics.

Despite the prevalence of mimics (Elde and Malik, 2009; Lasso et al, 2021; Boys et al, 2023; Senkevich et al, 2021; Mutz et al, 2023; Baptista et al, 2025), their importance for host modulation (Schönrich et al, 2017; Angulo et al, 2019; Kim and Weitzman, 2022; Ouyang et al, 2014; Graham et al, 2008) and potential roles in determining viral host range (Rothenburg and Brennan, 2020; Haller et al, 2014), central questions regarding the evolution of mimicry remain open. Two mimicry systems whose evolution was explored involved mimics that act as competitive inhibitors to antagonize host cellular response against viruses: (1) disruption of

blocking translation by eIF2α (eukaryotic Initiation Factor 2 alpha) mimicry (Elde et al, 2009; Jacquet et al, 2022; Chambers et al, 2024; Rothenburg et al, 2009), and (2) interference with cell death induction by viral mimics of MLKL (Mixed Lineage Kinase domain Like pseudokinase)(Petrie et al, 2019; Palmer et al, 2021). Both systems provided important insights into the evolution of mimicked-, mimicking-, and mutual target proteins, but also pointed to complex and diverse evolutionary patterns. Furthermore, since both systems include host proteins involved in immunity, the observed rapid evolution in some of their proteins may not represent other host-mimicked and target proteins. Indeed, many host-mimicked and target proteins are not involved in antiviral defense, and may evolve under different functional and regulatory constraints. These important but sporadic findings call for comprehensive analyses of mimicry systems and their evolution.

Here, we combine structural modeling and structural homology tools, human–virus and within-human PPI maps, with phylogeny-based approaches to infer evolutionary rates and detect selection in host and viral proteins. These allow us to characterize the evolutionary dynamics of viral-mimicking, host-mimicked, and target proteins and their interface residues, to compare between perfect and imperfect mimicry systems, to disentangle the evolutionary patterns of different interface regions, and to link the observed evolutionary trends with potential functional and cellular characteristics that may constrain host evolution. Our analysis shows that an evolutionary arms race is rarely found in mimicry-related host–virus interactions, and instead suggests that evolutionary conservation, driven by various constraints placed on the interacting partners and their interface regions, is far more prevalent.

## Results

### Detecting viral domain mimics and their host targets

We focused on five large dsDNA viruses: vaccinia virus (VACV), the type species of a genus of mammal-infecting poxviruses, and four human-infecting viruses, belonging to the *Orthoherpesviridae* family: herpes simplex 1 (HSV1), Epstein–Barr virus (EBV), human cytomegalovirus (HCMV), and Kaposi's sarcoma-associated herpesvirus (KSHV). These viruses are known to encode numerous host-like domains (Senkevich et al, 2021; Haller et al, 2014; Schönrich et al, 2017; Ouyang et al, 2014; Kvansakul et al, 2017) and have experimentally characterized human–virus PPIs (Yang et al, 2021; Davis et al, 2015; Nobre et al, 2019), allowing us to detect mutual host targets—the human proteins that both the viral mimics and the host-mimicked proteins interact with. Furthermore, these viruses have diverse tissue and cell tropisms and differ in replication mechanisms, providing us with a set of diversified viruses.

To find host-like domains in viral proteomes, we used AlphaFold2 (Tunyasuvunakool et al, 2021), and predicted the structures of 540 viral proteins (from the proteomes of the five viruses used in the analysis, downloaded from Uniprot—see additional details in "Methods"). The list of all viral proteins and their accession IDs appears as Dataset EV1. We then employed Foldseek(Barrio-Hernandez et al, 2023), a new structural homology method, to identify viral proteins that structurally resemble host

domains in the human proteome (see Fig. 1B for details on pipeline, and Fig. 1C for an example of host-mimicked — viral-mimicking pair). We then filtered matching 33,011 human-viral structural homologs, using a series of filtering stages, to remove spurious matches, including matches between low-complexity folds and between structures that differ significantly in most parts of the domain (see "Methods" for details). This series of filtering stages was benchmarked against a set of known host-like domains from different viruses and folds. Overall, we obtained 7593 viral-human structural homolog pairs, from 144 viral proteins. The large number of structural homolog pairs stems from the fact that some viral protein domains match more than a single human protein, with some of the human mimicked-domains being highly abundant (for example, similarities to ankyrin repeats or to immunoglobulin domains result in many homologs). The detected viral domain mimics (represented by the pink protein in Fig. 1A) comprise a significant fraction of the analyzed viral proteomes, ranging between 11 and 31% of the overall proteome in HSV1 and VACV, respectively. This suggests that a significant proportion of the proteins encoded by these large dsDNA viruses has maintained high similarity to host domain structures from which they presumably originated, and in line with previous results (Mutz et al, 2023; Senkevich et al, 2021; Koonin et al, 2022; Lasso et al, 2021). We refer to these viral proteins that include a host-like domain as "viral mimicking proteins", or viral mimics. We refer to their human homologs as "structural mimicked". The complete list of structurally mimicked human proteins and their viral-mimicking homologs is found in Dataset EV2, with details regarding their similarity in sequence and structure, and Dataset EV3 includes all human proteins and the categories they belong to in this study. Table EV1 includes general statistics on various groups of host and viral proteins used and analyzed in this work. We note that despite the structural similarity between host-mimicked and viral mimicking domains, their sequence similarity is often low (an average of 28.4% and 6.3% similarity between the mimicked and mimicking proteins in the homologous region and across all protein sequences, respectively). Thus, viral-mimicking domains are regarded as imperfect mimics, due to differences in the sequence and potentially smaller differences in structure.

Next, we asked which of the structural homologs represent pairs of host-mimicked and viral-mimicking proteins that function in a similar manner. In the context of this analysis, of host and viral interactomes, functional similarity refers to the host and viral homologs interacting with at least one mutual host protein. Thus, from the large lists of pairs of structural homologs, we searched for those pairs that have at least one shared interaction with a human protein (Fig. 1D). For this, we used human–virus and within-human PPI networks from existing databases (Yang et al, 2021; Szklarczyk et al, 2023), to find which viral-mimicking proteins interact with the same human protein as their human-mimicked homologs. In other words, we identified human proteins known to interact with both the host-mimicked and the viral-mimicking proteins. We refer to these human proteins as "mutual host targets" (see schematic representation in Fig. 1A, and Fig. 1E for an example of structurally resolved protein complexes of host-mimicked and viral-mimicking proteins with their mutual host target). Overall, we identified 645 mimicked-mimicking-target triads, out of 7593 homologous pairs. These include 229 human-mimicked and 40 viral-mimicking proteins, with 118 mutual host targets (see full list

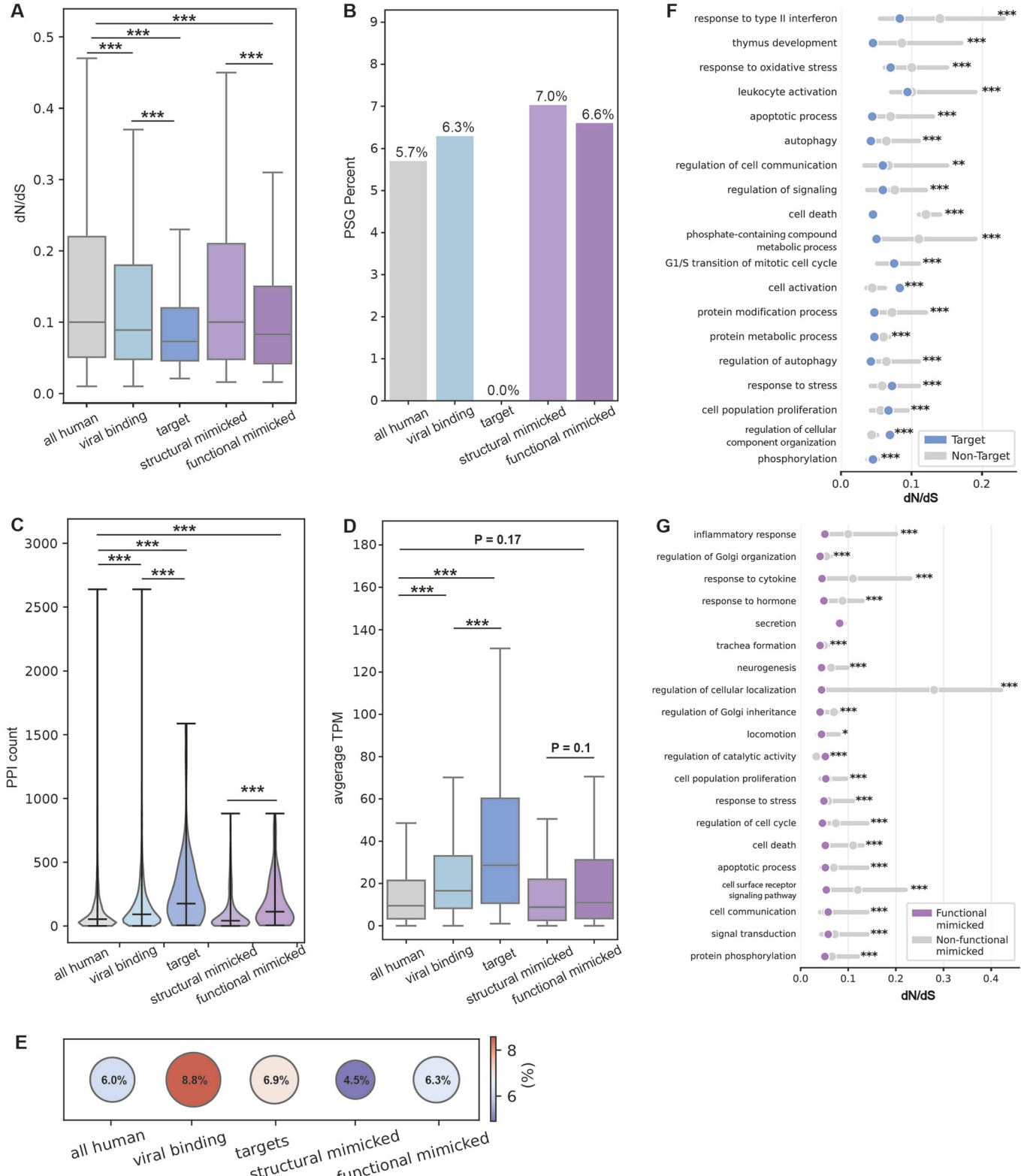

**Figure 2. Gene-level evolutionary analysis of host targets and mimicked proteins.**

(A) Boxplots showing the distribution of evolutionary rates of different human protein sets: all human proteins (9178 proteins), human proteins experimentally known to interact with at least one viral protein from the set of the five dsDNA viruses used in this study (viral-binding, 3489 proteins), human proteins known to interact with viral mimics and host-mimicked proteins (mutual targets, 58 proteins), human proteins that have at least one structural homolog in the five viral proteomes used in this study (structural mimicked, 586 proteins), a subset of structural mimicked that are also known to interact with the same target as viral mimicking proteins (functional mimicked, 106 proteins). The evolutionary rate is relative across all protein residues (5,572,013 residues in total) and is computed based on substitutions across a set of 10 one-to-one orthologs in primates, using the ratio between the number of non-synonymous and synonymous substitutions (dN/dS values, left). Groups were compared using Mann–Whitney test and corrected by FDR. $P$ values for comparisons of all vs. viral binding, target, functional mimicked, viral-binding vs. target, and structural vs. functional mimicked are 0.0, 0.0, 0.0, 3.31e-280, 0.0, respectively. (B) Bar plots showing percentage of genes within each group described in (A), with signatures of positive selection (PSGs). PSGs were identified based on a likelihood ratio test between two models, and based on a statistical significance threshold. Percentage of PSGs are shown with FDR-corrected $P$ value of 0.01 (additional thresholds are shown in Fig. EV1). Statistical enrichment (or depletion) was computed for each gene subset with respect to the group of all human genes using Fisher's exact test and corrected by FDR. None of the comparisons resulted in a significant depletion (i.e., all $P$ values > 0.05). Total (and positively selected) gene numbers are: all human – 8664 (494), viral binding—3559 (224), target—55 (0), structural mimicked—583 (41), functional mimicked—106 (7). (C) Violin plots showing the number of within-host PPIs for the sets of proteins defined in (A) (each filled area extends to represent the entire data range). $P$ values for the groups mentioned in (A), are 2.365e-86, 1.31e-11, 3.95e-09, 7.91e-06, 1.328e-10, respectively. (D) Boxplots showing the distributions of average gene-expression levels across healthy adult human tissues for each gene group described in (A). $P$ values for the groups mentioned in (A), are 5.5e-125, 8.1e-09, 1.7e-001, 5.0e-03, 1.0e-01, respectively. (E) Dotplots showing the fraction of essential genes within each set of genes defined in (A). Statistical enrichment (or depletion) was computed for each gene subset with respect to the group of all human genes using Fisher's exact test and corrected by FDR. None of the comparisons resulted significant, except for viral-binding versus all human proteome ($P$ value $= 5.4 \times 10^{-8}$). (F) Evolutionary rates of targets versus non-targets in a non-redundant list of enriched GO terms. (G) Evolutionary rates of functional-mimicked versus non-mimicked in a non-redundant list of enriched GO terms. (F, G) Groups were compared using a Mann–Whitney test and corrected by FDR, complete lists of GO terms appear in Dataset EV5. In all panels: ***$P < 0.001$, **$P < 0.01$, *$P < 0.05$. Boxplots in (A, D) represent the median, first quartile, and third quartile with lines extending to the furthest value within 1.5 of the interquartile range (IQR). Source data are available online for this figure.

in Dataset EV2). We refer to the set of human structural mimicked proteins, which are also known to interact with the same human target as the viral mimicking protein, as "functional mimicked". Thus, from the set of 2117 human proteins that are structural mimicked, there are 229 experimentally validated functional-mimicked proteins. Since the set of functional-mimicked proteins is the most physiologically relevant set for our analysis, we mostly refer to it in the following analyses, using the set of structural-mimicked proteins as a control.

## Host functional-mimicked, and target proteins are highly conserved proteins

We next employed phylogeny-based approaches to study coding-sequence evolution of all human protein-coding genes across primates, to compare evolutionary trends between genes encoding for host-mimicked and mutual target proteins, and other human protein groups (Fig. 1F). For this, we obtained 10,569 one-to-one orthologs from 9 non-human primates, from the ENSEMBL COMPARA database (Herrero et al, 2016). For each set of orthologs, we then created MSAs (Multiple Sequence Alignments), and masked poorly aligned regions and recombination regions, removing MSAs with too many masked regions, resulting in a set of 9178 MSAs. We then employed Selecton (Stern et al, 2007) to compute dN/dS—the ratio between nonsynonymous and synonymous substitutions—a measure indicative of the selective pressure on coding sequences, for each gene and for each residue within the genes in our dataset.

We compared evolutionary rates between different groups of human proteins: all human proteins, the subset of human proteins identified to be mimicked by the five dsDNA viruses we used in this study, including structural mimicked and the functional mimicked, the set of human proteins known to be experimentally interacting with viral proteins encoded by these five dsDNA viruses, and those human proteins that are mutual targets of both host-mimicked and viral-mimicking proteins (Fig. 2A). We observe that viral-binding

human proteins are significantly more conserved than the overall set of human proteins, in line with previous findings (Shuler and Hagai, 2022; Davis et al, 2015; Hagai et al, 2018; Enard et al, 2016; Dyer et al, 2008). Mutual targets, those human proteins that interact with viral mimics and with their host-mimicked homologs, are the most conserved set of genes we tested, significantly more conserved than the overall set of human proteins, as well as the set of human proteins that interact with viral proteins. Finally, functional-mimicked proteins (the subset of structural homologs that have evidence of similar interactions with host targets as the viral-mimicking proteins) are significantly more conserved than the superset of all structural-mimicked proteins, and more conserved than the entire set of human proteins (Fig. 2A). In summary, both the functional-mimicked and mutual-target proteins undergo significantly slower evolution than the rest of the human proteome and several control groups, across primates.

## Few host targets of viral mimics have evidence of pervasive positive selection

We next asked whether the various groups of proteins we study differ in frequencies of proteins with signatures of pervasive positive selection (PSGs, positively selected genes). For this, we employed the *codeml* program from PAML (Yang, 2007) and compared two site-models (M8 versus M8a) using a likelihood ratio test to detect proteins that undergo positive selection across primates (Figs. 2B and EV1). While the groups tested differ in their fraction of PSGs, none is significantly depleted or enriched with respect to the set of all human proteins, likely due to the small numbers of PSGs in each group (ranging from 0 to 494, in target and all proteins, respectively). The group of viral-binding human proteins has a higher fraction of proteins with signatures of positive selection than the overall group of all human proteins (6.3% versus 5.7%, respectively), in line with previous results that used different models, evolutionary timescales and sets of host proteins (Cariou et al, 2022; Enard et al, 2016). Proteins targeted by mimics,

however, have no detected PSGs in most confidence levels (Figs. 2B and EV1). Host-mimicked proteins have a slightly higher occurrence of positively selected proteins (7.0% and 6.6% in structural and functional mimicked, respectively) than all human proteins. The results shown in Fig. 2B are consistently observed with different stringency levels for PSG detection (Fig. EV1).

In summary, human proteins targeted by both host-mimicked and viral-mimicking proteins evolve slowly in the course of primate evolution, more so than other human proteins targeted by viruses. This protein group includes a lower fraction of proteins with signatures of positive selection than the entire set of human proteins, and also when compared to the group of viral-binding human proteins.

## Functional and cellular characteristics of mimicked and target proteins suggest functional constraints imposed on these host proteins

Following these results of higher conservation of host proteins related to mimicry, we asked whether various cellular and functional characteristics may be related to this conservation. We began by comparing the number of PPIs formed between proteins from the groups we study and other human proteins. For this, we used the number of physical interactions that each human protein is experimentally known to form with other human proteins, taken from STRING (Szklarczyk et al, 2023) (Fig. 2C; see Dataset EV4 for a list of human proteins, their PPI numbers and additional cellular characteristics). Essential interactions between host proteins and other host proteins may place constraints on the interacting proteins, and in particular on their interface regions. We observe that the group of viral-binding human proteins has significantly higher numbers of interactors than the whole human proteome (FDR-corrected $P$ value $= 1.9 \times 10^{-86}$). The group of human proteins targeted by viral mimics has a distribution with even higher numbers of interactors, significantly higher than the overall set of human proteins that interact with viral proteins through various mechanisms (FDR-corrected $P$ value $= 9.4 \times 10^{-5}$ when comparing host targets with viral-binding human proteins). Finally, functional-mimicked proteins tend to have higher numbers of interactors than structural-mimicked proteins, and in comparison with the set of all human proteins.

We next compared the distributions of gene expression levels between the various groups, since gene expression level is known to be a major determinant of gene evolutionary rate (Drummond et al, 2005; Echave and Wilke, 2017). For this, we used two different metrics—the average TPM (transcript per million) per gene across tissues, and the number of tissues that express this gene, using bulk RNA-seq data from healthy tissues from the Human Protein Atlas (Uhlén et al, 2015) (Fig. 2D; Appendix Fig. S1). We observe that viral-binding proteins are more highly expressed than the rest of the human proteome. Furthermore, the subset of viral-binding proteins targeted by mimicry tend to be more highly expressed than other viral-binding proteins. Host-mimicked proteins do not show significantly different levels of expression than the overall set of human proteins or when comparing structural- and functional-mimicked proteins.

Finally, we used a dataset of genes known to be essential (Bartha et al, 2018), to test whether different groups in our analysis have higher proportions of essential genes (Fig. 2E). We observe that the

set of viral-binding proteins has the highest fraction of essential genes (8.8% with respect to 6.0% in the entire human proteome). Both functional-mimicked and target proteins have slightly higher fractions of essential genes than all human genes, however the differences between these groups are relatively small.

Next, we used GO term analysis to test which functions are enriched in host proteins related to mimicry. Both functional mimicked and targets are enriched with various antiviral pathways ("cell death", "response to cytokine", "inflammatory response") and in pathways often modulated by viruses ("regulation of Golgi", "G1/S transition of mitotic cell cycle", "signal transduction"). In each GO term category, we tested whether proteins related to mimicry are more conserved (Fig. 2F,G; Dataset EV5). We observed that functionally mimicked proteins are more conserved than non-mimicked proteins belonging to the same GO category in 158 out of 209 (75.6%) enriched terms. Targets are more conserved than non-targets in 36 out of 57 (63.2%) enriched terms. Thus, in the majority but not in all cases, mimicked and target proteins are more conserved than respective host proteins within the same functional category.

In summary, we observe that proteins involved in mimicry interactions, including functional-mimicked and target proteins, tend to interact with higher numbers of human proteins than the overall set of human proteins. Host targets of viral mimics tend to be more highly expressed than other human proteins, including other human proteins targeted by viral proteins through diverse mechanisms, unrelated to domain mimicry. These results suggest that host proteins involved in mimicry interactions are under stronger functional and cellular constraints to evolve. These constraints may limit host capacity to escape deleterious interactions with viral mimics. In the next sections, we will directly test this possibility by focusing on interface regions of host-mimicked and target proteins, and on their evolution.

## Interface residues of host-mimicked and target proteins are highly conserved

Mimicry poses a challenge for the host: the viral-mimicking protein uses the same interface as the host-mimicked protein to interact with their mutual target. While this interaction is beneficial for viral replication, it can be deleterious for the host. Thus, escaping these interactions with mimics through interface mutations may increase host fitness. In this scenario, an evolutionary arms race may ensue, as both the host and the virus strive to escape or maintain these interactions through mutational changes at the interface region (Daugherty and Malik, 2012; Duggal and Emerman, 2012). However, mutations at the interface that can weaken the interaction between the host target and the viral mimic, can also weaken the interaction between the host target and the host-mimicked protein, that uses the same interface, perturbing the cellular interactome. We thus asked which evolutionary patterns and trends are observed at interface regions between host-mimicked and target proteins.

To identify interface regions between host-mimicked and target proteins, we employed AlphaFold-multimer (Evans et al, 2021) with 576 mimicked-target pairs and used the resulting 59 protein complexes whose prediction scores were sufficiently high ($0.8 \times ipTM + 0.2 \times pTM > 0.7$). We also used 12 complexes of viral-mimicking and host–target proteins with high-enough prediction

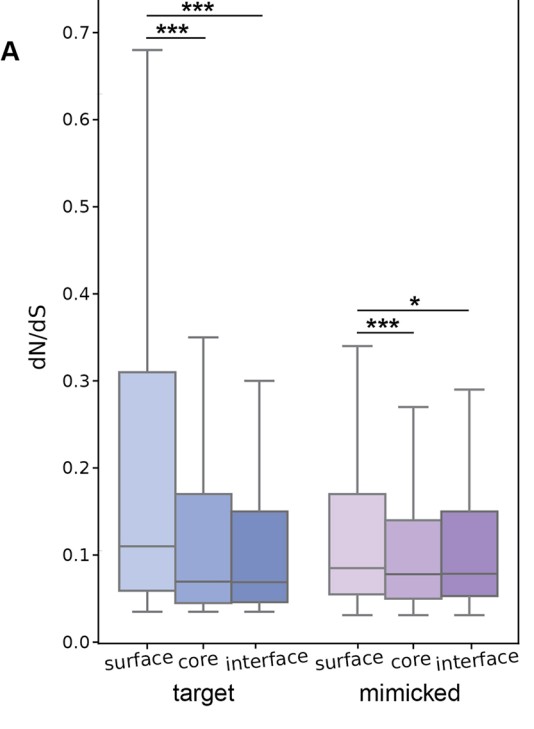

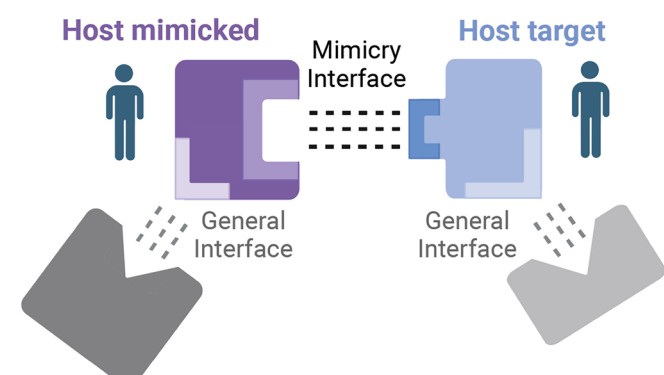

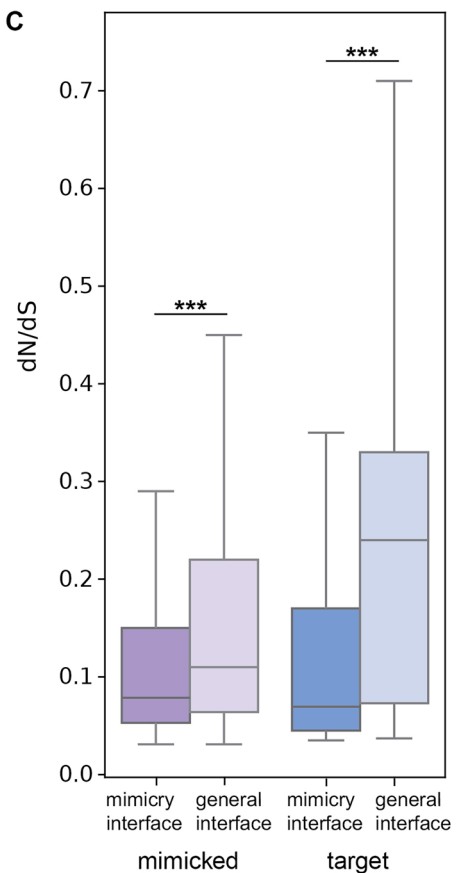

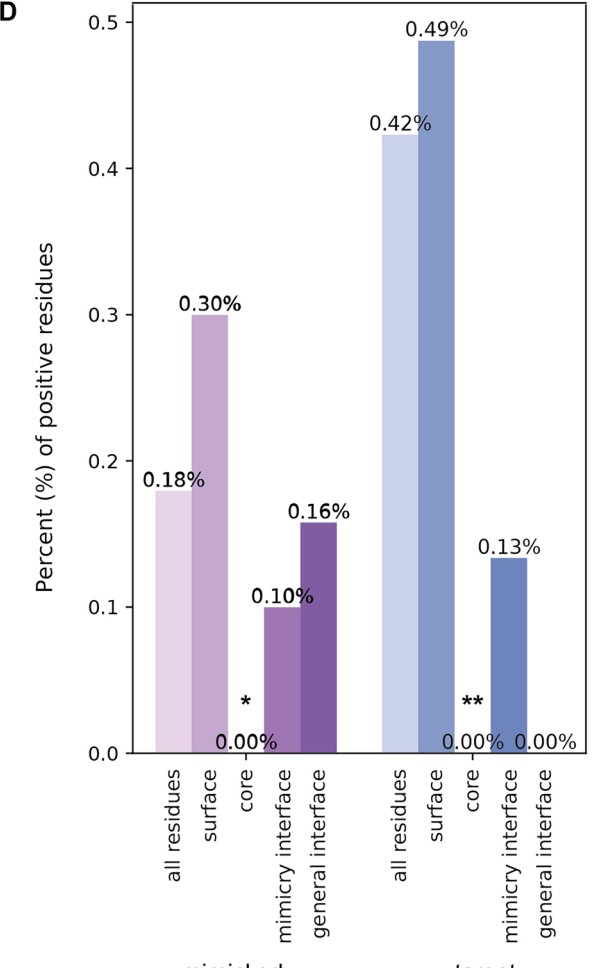

**Figure 3.   Interface-level evolutionary analysis of host targets and mimicked proteins.**

(A) Evolutionary rates of core, surface, and interface residues in host targets and mimicked proteins (based on 59 predicted protein complexes of which 12 mimicked and 15 target proteins have dN/dS values, resulting in 3810 and 5,012 analyzed residues in mimicked and target proteins, respectively). Interface residues are defined as those that mediate interactions between host-mimicked or viral-mimicking and target proteins. Core, surface and residues are identified using the contacts of structural units (CSU) method. Groups were compared using Mann–Whitney test and corrected by FDR. $P$ values for comparisons of target surface vs. core, interface and mimicked surface vs. core, interface are 2.06e-51, 9.56e-26, 6.653e-07, 1.4e-02, respectively. $N$ of residues in mimicked - surface, core, interface and target - surface, core, interface are 1603, 1565, 642, 2557, 1693, 839 residues, respectively. (B) A schematic illustration showing two types of interface regions – those involved in interactions with mimicked proteins (mimicry interface) and those involved with interacting unrelated proteins (general interface). (C) Comparison of evolutionary rates between mimicry interface and general interface residues, in mimicked and target proteins. Groups were compared using Mann–Whitney test and corrected by FDR. General interface was predicted using ScanNet, taking residues with a score of at least 0.7. $P$ values for comparisons of mimicked general interface vs. mimicked mimicry interface, and target general interface vs. target mimicry interface are 8.85e-10 and 1.80e-37, respectively. $N$ of residues in mimicked - general interface, mimicry interface and target - general interface, mimicry interface are 371,642,562,762 residues, respectively. (D) Percentage of residues with signatures of positive selection (%PSR), in mimicked and target proteins, in each of the following groups of residues: all residues, surface, core, mimicry interface and general interface. Statistical enrichment (or depletion) was computed for each gene subset with respect to the group of all human genes using Fisher's exact test and corrected by FDR. None of the comparisons resulted in a significant depletion (i.e., all $P$ values > 0.05) except for core residues in both host-mimicked and target proteins. $P$ values for comparisons of mimicked all residues vs. surface, core, mimicry interface, general interface vs and target all residues vs. surface, core, mimicry interface, general interface are 0.0389, 0.0999, 0.5899, 0.6829, 0.008, 0.349, 0.295, 0.349, respectively. In all panels: ***$P$ < 0.001, **$P$ < 0.01, *$P$ < 0.05. Boxplots in (A, C) represent the median, first quartile and third quartile with lines extending to the furthest value within 1.5 of the interquartile range (IQR). Source data are available online for this figure.

scores (out of 116 pairs), to similarly identify interface regions between viral mimics and host target proteins. We then identified the interface residues between these proteins, defined as residues that form direct interactions with residues from the other protein within the complex, using the contacts of structural units (CSU) method (Sobolev et al, 1999). And, using the same program, defined all other residues as either surface-exposed or core residues. We observe that in both mimicked and target proteins, the interface residues evolve significantly more slowly than surface residues, as expected by constraints to maintain these interactions. Core residues also display higher conservation in comparison to surface residues (Fig. 3A).

## Mimicry-related interface regions are more conserved than other interface regions and show little evidence of positive selection

We next identified residues predicted to participate in interactions with other proteins, unrelatedly to the mimicry system's interactions, in both host-mimicked and target proteins. This was done using two methods for predicting binding sites on protein surfaces, based on the protein monomeric structure: ScanNet (Tubiana et al, 2022), a recently developed program that predicts binding sites on protein surfaces, based on a geometric deep learning model that uses the spatio-chemical arrangement of amino acids and their neighbors, and ISPRED4 (Savojardo et al, 2017), that combines Support Vector Machines with Conditional Random Fields (CRF) to predict interface sites.

Thus, we obtained two types of interface residues: (1) those involved in mimicry interactions (residues in the mimicked protein that interact with the target protein, and the residues at the target–protein surface that interact with host-mimicked or with viral-mimicking protein residues, identified based on the mimicked-target protein complex structures) and (2) residues involved in interacting with other proteins, unrelated to the mimicking- or mimicked-target interactions (predicted using either ScanNet or ISPRED4). We term these "mimicry-interface" and "general-interface" residues, respectively (see schematic representation of both interface types in Fig. 3B). We observe that in both mimicked and target proteins the mimicry-interface residues are

significantly more conserved than the general-interface residues (Fig. 3C). The same trends were observed for both mimicked and target proteins, when using different parameters to define "general-interface" residues, based on different cutoffs of the ScanNet prediction score, based on exclusion or inclusion of disordered regions from these calculations, and based on ISPRED4 predictions (Fig. EV2). This suggests that interface regions involved in interactions with mimicry are under stronger purifying selection than those interface regions mediating other interactions.

We next asked what fraction of interface residues evolved under positive selection and whether this fraction of positively selected residues (PSRs) is relatively high in comparison to PSRs in other protein regions. For this, we used the structural partition of residues of both target and mimicked proteins into core, surface, mimicry-interface, and general-interface, as described above. We observe that in both target and mimicked proteins, surface residues have higher fractions of PSRs than the set of all residues, and core residues are significantly depleted of PSRs, as expected (Fig. 3D). Both types of interface regions have lower fractions of PSRs than the overall set of residues, and in comparison with surface residues (although these differences are not statistically significant). In mimicked proteins, the general interface has a slightly higher fraction of PSRs than the mimicry interface (0.16% (of 634 residues) versus 0.1% (of 1002 residues), respectively), while in target proteins only a single PSR is found in mimicry-interface regions (0.13% (of 749 residues)). These results suggest that interface regions, in both mimicked and target proteins, have lower occurrence of PSRs, and do not support a scenario of rapid evolution in these interface regions in the majority of host proteins involved with viral mimicry.

## Interface residues targeted by both host-mimicked and viral-mimicking proteins are more conserved than residues targeted only by viral-mimicking proteins

Domain mimics are thought to be acquired from the host through various mechanisms (Rahman et al, 2022; Fixsen et al, 2022). However, these acquired host genes rarely remain unchanged and often diverge in sequence from the original host copy, as observed even in cases of relatively recent acquisitions (Martínez-Vicente

et al, 2020; Gubser et al, 2007; De Sabato et al, 2020). Thus, viral domain mimics are regarded as imperfect mimics, and this can lead to differences between the residues at the surface of host targets that interact with the host-mimicked and the viral-mimicking proteins. Residues at the surface of host targets that only interact with viral-mimicking proteins and not with host-mimicked proteins, may provide a basis for the host to discriminate between host- and viral-interacting proteins, allowing for potential escape from deleterious interactions with viral proteins through changes in these specific residues. We thus asked whether different residues within the mimicry-interface of the host–target protein display different evolutionary patterns, based on whether they interact with the host-mimicked protein, the viral-mimicking protein, or both.

We used the predicted protein complexes of viral-mimicking—host–target complexes (12 complexes with prediction scores >0.7), in addition to the respective host-mimicked—host–target complexes that share the same mutual targets (59 complexes with AlphaFold prediction scores >0.7). We thus obtained pairs of homologous protein complexes, one with the viral mimic and another with the host-mimicked protein, both interacting with the same host target. We used these protein complex structures to detect interface residues that (a) only interact with the host-mimicked protein, (b) only interact with the viral-mimicking protein, or (c) interact with both. These are referred to as "hostspecific", "virus-specific", and "mutual" residues, respectively (see schematic diagram in Fig. 4A). We then used the evolutionary rates of these residues, computed as described above, to compare these three different interface regions. We observe that the mutual-interface residues are the most conserved residues within the interface (Fig. 4B). Furthermore, interface residues interacting with only the viral mimicking protein are the fastest evolving among the three groups.

In summary, interface residues involved in mimicry interactions tend to be more conserved than other residues used for interaction with other host proteins, unrelated to mimicry (as observed in Fig. 3C). Within the interface region in target proteins, residues that interact with both host-mimicked and viral mimicking proteins are more conserved than those that interact only with the viral protein (Fig. 4B). The observation that residues in host surfaces that exclusively interact with viral mimics display rapid evolution may suggest that fewer constraints are imposed on these residues, since they are not essential for interactions with host proteins. It may also suggest that these regions, where only viral proteins interact with the host, can provide a basis for the host to escape these viral interactions without compromising interactions with host proteins. However, we note that these regions are not enriched with PSRs (Fig. 3D). Positive selection was previously used as a basis for detecting regions that potentially evolve following antagonistic interactions between host and viruses, and our analysis of mimicry-related interfaces did not detect many such cases of positive selection. We further discuss these results and their potential interpretation in "Discussion".

## Host proteins targeted by perfect mimicry are more conserved and subject to greater constraints than host proteins targeted by imperfect mimicry

As shown above, imperfect mimicry of domains leads to differences in interactions between the host-mimicked and the viral-mimicking

proteins, and the regions exclusively targeted by each may display different evolutionary patterns. This led us to ask whether there are differences, at the global gene-level, in evolutionary rates between host proteins targeted by perfect mimics and those targeted by imperfect mimics. It was previously suggested that perfect mimics would target highly conserved host targets (Elde and Malik, 2009). This is because the host is unable to discriminate between perfect mimics and mimicked proteins, unlike the above analysis that concerned imperfect domain mimics. To obtain a large set of host proteins targeted by perfect mimics, we chose to focus on host–virus PPIs mediated through domain-motif interactions. In these cases, short linear motifs embedded in disordered regions of viral proteins perfectly mimic analogous motifs found in host proteins, to form interactions with specific host domains (e.g., SH3-binding motifs specifically interact with SH3 domains) (Davey et al, 2011; Glavina et al, 2022; Via et al, 2015). These motif mimics were suggested to occur in viral proteins from a large and diverse set of viruses, including in the set of dsDNA viruses we used in this study (Hagai et al, 2014; Garamszegi et al, 2013; Halehalli and Nagarajaram, 2015; Mihalič et al, 2023).

We employed a computational approach to detect motifs in viral proteins, their host-mimicked motifs, and mutual host targets, using existing human–virus and within-human PPI datasets (see "Methods" for details). This has yielded 89 host proteins that are inferred to be targeted by viral motifs. To distinguish between various host targets used in this analysis, we term these "motif-mimicry targets", while the targets analyzed in the previous sections are here referred to as "domain-mimicry targets" (Fig. 4C). We observe that motif-mimicry targets are more conserved than domain-mimicry targets and also than other viral-binding host proteins, comprising the most conserved set of proteins tested in this study (Fig. 4D). We next compared cellular and functional characteristics between motif- and domain-mimicry targets, the overall set of all human proteins targeted by viral proteins and all human proteins. We observe that motif-mimicry targets tend to have the highest number of PPIs within the human interactome and to be the most highly expressed, followed by domain-mimicry targets (Fig. EV3). We also observe that the group of viral-binding human proteins has the highest fraction of essential genes among the tested groups (Fig. EV3).

These results suggest that targets of perfect mimicry, in the form of short linear host-like motifs, are placed under strong constraints, including higher expression levels and higher numbers of interactions than observed in other tested groups of human proteins. These constraints limit host capacity to escape interactions with viral motifs. This is especially true given the fact that perfect mimicry of host motifs forms interactions with host domains in an identical fashion to that formed between the mimicked host motifs and those domains.

## Viral mimics show complex patterns of conservation, with few genes displaying pervasive signatures of positive selection

Following the evolutionary analysis of host-mimicked and target proteins, their interface regions and constraints, we next focused on viral mimicking proteins and their evolutionary patterns. We obtained one-to-one orthologs, of simplexviruses, cytomegaloviruses, rhadinoviruses, and orthopoxviruses, for the four human-

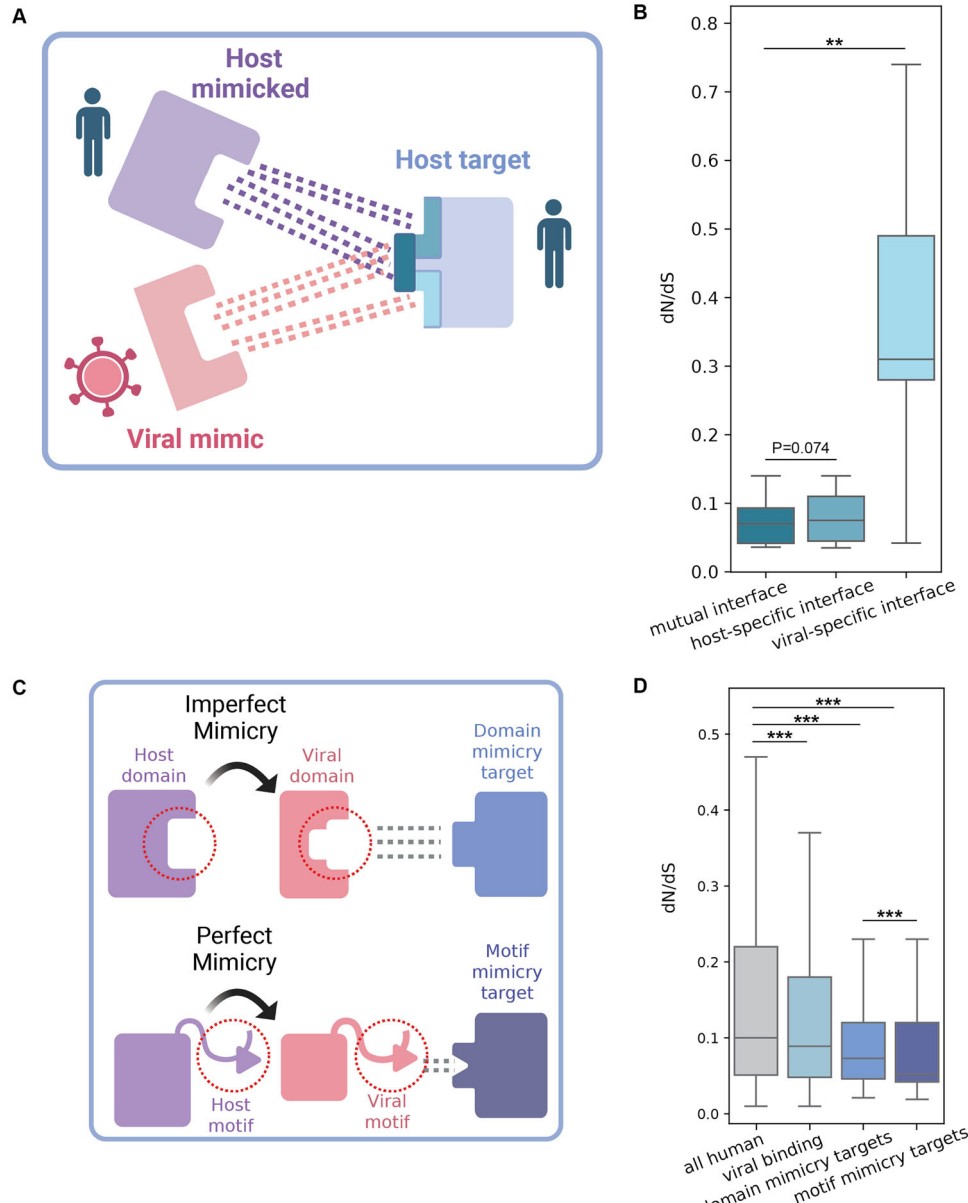

**Figure 4. Evolutionary comparison between interface regions and between perfect and imperfect mimicry targets.**

(A) A schematic illustration showing the interface residues in the host target proteins that interact with both host-mimicked and viral mimicking proteins (mutual interface residues), or that interact with only one of them (host-specific and viral-specific interface residues). (B) Comparison of evolutionary rates between interface residues belonging to mutual interface, host-specific or viral-specific regions (based on 19 triads of host-mimicked, viral-mimicking and mutual targets, 38 complexes in total). Groups were compared using Mann–Whitney test and corrected by FDR. *P* values for comparisons of mutual-interface vs. host-specific interface, viral-specific interface are 0.074 and 0.005, respectively. (C) Schematics of host proteins targeted by imperfect domain mimics (top) and perfect motif mimics (bottom). (D) Comparison of host proteins targeted by perfect and imperfect viral mimics. Evolutionary rates of different human groups, as shown in Fig. 2A, with the addition of a group of human proteins that interact with perfect mimics of linear motifs through domain-motif interactions (49 proteins). Groups were compared using Mann–Whitney test and corrected by FDR. In all panels: ***P < 0.001, **P < 0.01, *P < 0.05. Boxplots in (B, D) represent the median, first quartile and third quartile with lines extending to the furthest value within 1.5 of the interquartile range (IQR). *P* values for comparisons of all vs. viral binding, domain mimicry targets, motif mimicry targets, and domain mimicry targets vs. motif mimicry targets are 0.0, 0.0, 0.0, 1.889e-280, respectively. /ppImages were created in BioRender. (2025) https://BioRender.com/1m65sw7. Source data are available online for this figure.

infecting viruses we used in this study. For EBV, the fifth virus in the above analyses, there is an insufficient number of closely related viruses to carry out evolutionary analyses across orthologs, and it was thus excluded from this analysis. We employed phylogeny-based approaches similar to the primate analysis, based on PAML

(Yang, 2007) and Selecton (Stern et al, 2007) programs, to infer evolutionary rates of viral proteins and residues, and to identify sites with signatures of pervasive positive selection. Viral mimics may be under strong selection to maintain similarity to host-mimicked proteins, to retain interactions with host targets that are

formed in a similar manner to that used by host-mimicked proteins (Elde and Malik, 2009). On the other hand, interactions with host proteins may lead to an evolutionary arms race, as suggested in at least one case of domain-mimicking proteins (Elde et al, 2009). The evolutionary patterns of viral mimics were not previously studied across a large set of mimics, and we thus set out to study these trends.

At the protein level, when comparing the evolutionary rates of structural and functional viral-mimicking proteins with all viral proteins, we did not observe consistent patterns across the four viruses (Fig. EV4). These results can be explained by a range of technical factors, including a small dynamic range of dN/dS, few orthologous viral proteins (limiting rate variability between proteins and between residues), and a significantly smaller set of proteins with respect to the host protein analysis (a few dozen proteins in the viral analysis versus hundreds to thousands in the host analysis). Several biological factors may be behind these results as well, including the fact that viral proteins are often multi-functional. These multiple different functions may place constraints on viral proteins, in addition to the specific requirement placed on viral mimics to resemble host-mimicked proteins to interact with host targets. Furthermore, other structural determinants may influence viral protein evolutionary rates, such as fold complexity and the fraction of disordered regions, characteristics that are not directly related to mimicry but may have a strong influence on evolutionary rates (Fuchs et al, 2025). We tested whether mimics differ from non-mimics in various structural and functional characteristics thought to be related to evolutionary rates of these viral proteins (Fuchs et al, 2025; Molteni et al, 2023) (Fig EV5; Appendix Fig S2). However, none of these comparisons supported the notion that the evolutionary rates of mimics are biased by these characteristics in a consistent manner across all studied viruses.

We next asked whether viral-mimicking proteins differ in the fraction of proteins displaying positive selection (fraction of PSGs). Using site-models, we observed that in orthopoxviruses, a higher fraction of PSG occurrence in non-mimicking proteins in comparison with mimicking proteins (Fig. 5A, see Appendix Fig. S3 for %PSG analysis using additional threshold values). In herpesviruses, we did not detect any PSG, with the exception of a single KSHV protein that is a viral mimicking protein. Because of this low PSG occurrence in herpesviruses using site-models, we asked whether we can detect higher levels of PSGs using branch-site models to test for episodic positive selection in the branch leading to the human-infecting viruses. We found higher numbers of PSGs using branch-site models in all three tested herpesviruses, HSV1, HCMV, and KSHV, than the respective site-model analysis. In all three herpesviruses, there was a higher fraction of PSGs in non-mimicking than in mimicking proteins using branch-site models (Fig. 5B–D), similar to the observation in orthopoxviruses (Fig. 5A). This is true in all threshold values we used to detect PSGs in these four viruses, except for one case in KSHV analysis (where using an FDR-corrected $P$ value below 0.05 leads to detection of a single mimicking protein, Appendix Fig S3). We note, however, that these lower fractions of PSGs do not represent a significant depletion of PSGs in viral-mimicking versus all other viral proteins (see details in Fig. 5A–D). Instead, this lower level of PSGs in viral-mimicking proteins does not support the notion that viral mimicking proteins tend to evolve through positive selection, and are not enriched with PSGs in comparison with non-mimicking viral proteins.

## Interfaces of viral mimics are conserved and have few positively selected sites

Finally, we analyzed residue-level evolutionary rates in viral-mimicking proteins. For this, we used the host–virus protein complexes predicted with high scores in AlphaFold-Multimer (Evans et al, 2021). These included complexes formed between VACV and KSHV and human proteins (no complexes with HSV1 or HCMV proteins yielded predictions with high-enough scores). Using these host–virus protein complex structures, we identified core, surface and interface residues. When looking at the evolutionary rates of residues from these three regions, we observe that core residues are more conserved than surface residues, as expected (Fig. 5E,F). Interface residues show intermediate values between core and surface, and are significantly more conserved than surface residues. These results mirror our observations of rates of different regions in host-mimicked and target proteins (Fig. 3A). These results suggest that the majority of viral interface residues are conserved and are likely constrained due to their interactions with host target residues.

We next asked whether interface residues of viral mimicking proteins display signatures of positive selection. Only one of the viral proteins that had a predicted host-viral protein complex structure, the VACV Bcl2-like OPG035 protein, had one such positively selected residue. This result is expected, given the low number of PSGs in our viral-mimicking protein dataset. This residue is not part of the interface between the viral mimicking and the host target proteins. Instead, it is found at the surface of the viral protein, in a distant position with respect to the interface formed with the host target protein (Fig. 5G).

In summary, these analyses, while based on a relatively small number of proteins, do not provide evidence of rapid evolution or enrichment in positively selected sites in interface regions of viral mimics.

## Discussion

Conflicts arising from host–virus interactions should often lead to an evolutionary arms race where both sides evolve rapidly to counteract each other, since these interactions are selectively advantageous for one of the interacting partners but deleterious for the other. Indeed, it was shown that human genes that display signatures of positive selection tend to be enriched with antiviral functions or to encode for proteins involved in interactions with various viruses (Fumagalli et al, 2011; Sironi et al, 2015; Enard et al, 2016). Functional studies that tested how evolutionary changes in these positively selected host genes affect viral replication showed that these rapid changes across host species can significantly modulate the outcome of infection, implying that these signatures of positive selection are traces of evolutionary arms races between hosts and viruses. Importantly, these studies have shown that this evolutionary dynamic is observed in interactions involving both host restriction factors (host proteins that inhibit viral replication) (Daugherty and Malik, 2012; Duggal and Emerman, 2012; Tsu et al, 2021) and host dependency factors (host proteins that viruses hijack to assist in their replication)(Demogines et al, 2013; Meyerson et al, 2014). This dynamic is also observed in extracellular interactions, in the evolution of several surface

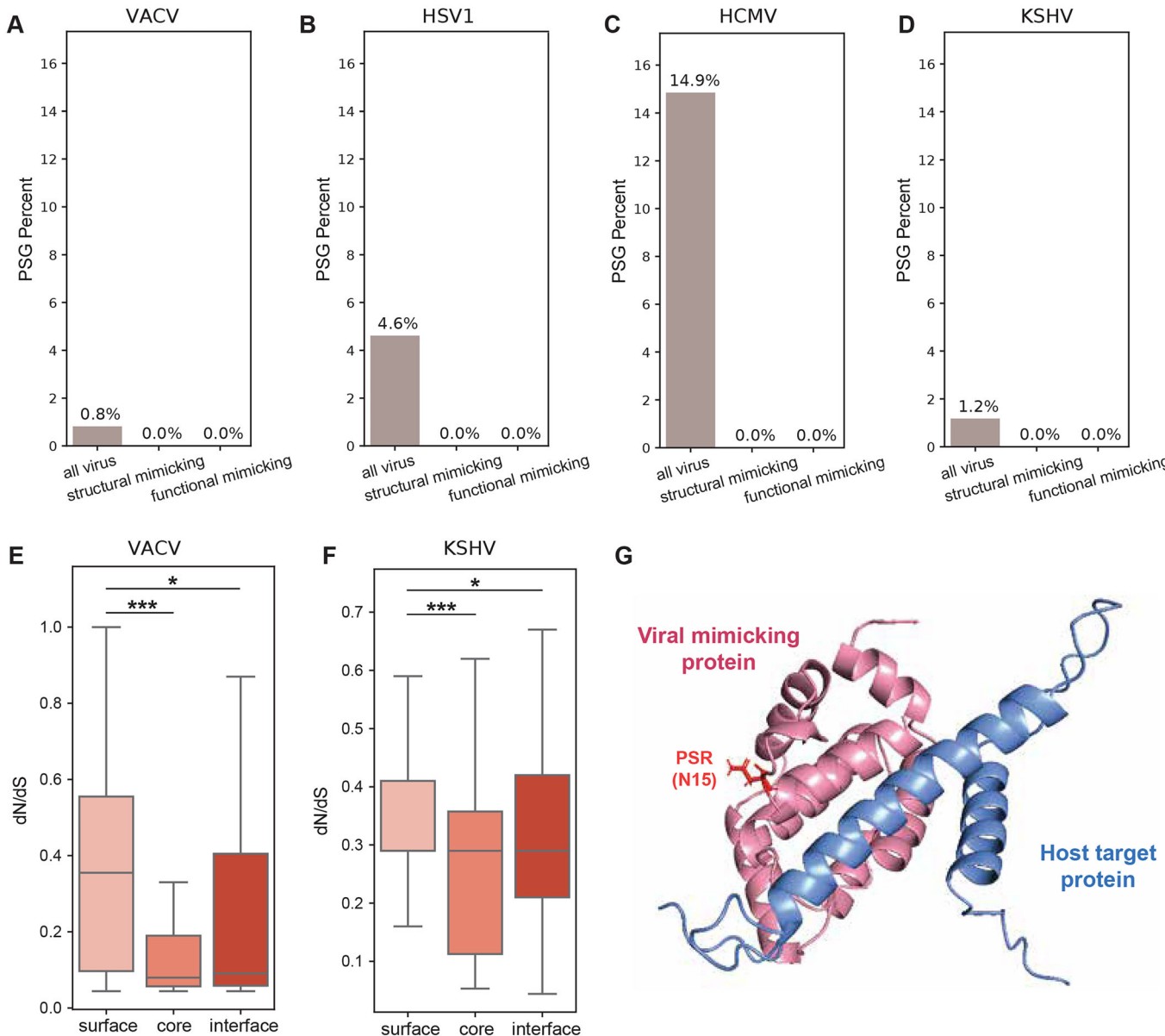

**Figure 5.  Evolutionary analysis of viral mimics and their interface regions.**

(A–D) Percentage of genes with signatures of positive selection (PSGs) in different viruses: orthopoxviruses, simplexviruses, cytomegaloviruses, and rhadinoviruses. Genes are partitioned into those encoding for functional and structural mimics (functional mimics are structural homologs of host domains that also have known mutual host targets with these host-mimicked proteins, while the group of structural mimics is all other structural homologs, with no known mutual targets) and compared with all viral coding genes. The percentage of PSGs are shown with an FDR-corrected *P* value of 0.01 (for additional *P* values, see Appendix Fig S3). Statistical significance for depletion of PSG fraction was computed for each gene subset with respect to the group of all human genes using Fisher's exact test and corrected by FDR. None of the comparisons resulted in a significant depletion (i.e., all *P* values > 0.05). PSGs were detected using site models (orthopoxviruses) and using branch-site models (all other viruses). (E, F) Evolutionary rates of core, surface and interface residues in viral-mimicking proteins from orthopoxviruses and rhadinoviruses (with structures taken from VACV and KSHV, respectively). Interface residues are defined as those that mediate interactions between viral-mimicking and host–target proteins. Core, surface and residues are identified using the contacts of structural units (CSU) method. Groups were compared using Mann–Whitney test and corrected by FDR. *P* values for comparisons of VACV surface vs. core, interface and KSHV surface vs. core, interface are 0.0011, 0.0237, 0.00063, 0.036, respectively. N of residues in VACV surface, core, interface and KSHV surface, core, interface are 42, 41, 30, 102, 150, 185 residues, respectively. (G) AlphaFold-Multimer structural prediction of VACV OPG035 Bcl2-like protein in complex with human BAD protein (in pink and blue, respectively). Residue N15 in OPG035, inferred to be positively selected across orthopoxviruses, is shown in red (marked with "PSR"). This residue is outside, and points in the opposite direction from, the interface with the human target protein. In all panels: ***P < 0.001, **P < 0.01, *P < 0.05. Boxplots in (E, F) represent the median, first quartile and third quartile with lines extending to the furthest value within 1.5 of the interquartile range (IQR). Structure images were created using PyMol (https://www.pymol.org/). Source data are available online for this figure.

receptors used for viral entry, and in antibody interactions with viral capsids (Kistler and Bedford, 2023; Starr et al, 2022; Jacquet et al, 2019; Warren et al, 2025).

While rapid evolution of host proteins may benefit the host in its ongoing conflicts with viruses, functional and cellular constraints imposed on host proteins may limit host capacity to evolve (Shuler and Hagai, 2022). This is especially true in host–virus interactions involving mimicry, since the mimic utilizes its structural similarity to interact with host targets in a similar manner to the interactions formed between host targets and host-mimicked proteins. Thus, hosts face a challenging scenario regarding (1) whether to evolve and escape mimics while compromising the cellular interactome and (2) how to discriminate between host-mimicked and viral-mimicking interactions.

Here, we investigated the evolutionary dynamics between a range of mimics encoded by different viruses and their hosts. We found that both host-mimicked proteins and targets of mimicry evolve more slowly than other human proteins. Human proteins targeted by mimics evolve more slowly than human proteins targeted by viral proteins that do not employ mimicry. Interestingly, human proteins targeted by perfect motif mimics evolve even more slowly than those targeted by imperfect domain mimics. Furthermore, host proteins mimicked by viruses or targeted by mimics rarely show evidence of positive selection, precluding the scenario that these proteins evolve rapidly as part of an evolutionary arms race with viruses. The conservation of these proteins may be explained by various cellular and functional constraints. Indeed, host-mimicked proteins and targets of both perfect and imperfect mimics are highly connected proteins and are expressed in relatively high levels across many tissues – characteristics that may limit their coding-sequence evolution.

At the interface level, interface residues involved in the mimicry system, of both host-mimicked and target proteins, are highly conserved across primates. These interfaces include few residues with signatures of positive selection. Instead, most PSRs in these proteins reside on surface regions, unrelated to interactions with mimicry. When comparing interface residues involved in mimicry interactions to those involved in interactions with other host proteins, mimicry-interface regions are significantly more conserved than other interface regions. This residue-level analysis further bolsters the notion that host proteins involved in mimicry systems tend not to engage in an evolutionary arms race with viral mimics. Importantly, in cases of imperfect mimicry, where the mimics and mimicked proteins may form slightly different interfaces with their host mutual targets, interface residues exclusively targeted by viral mimics tend to evolve more rapidly than those residues targeted by both host-mimicked and viral-mimicking proteins. This difference in evolutionary rates between sub-regions within the target interface may be related to constraints imposed on residues essential for interactions with host proteins, but not on residues uniquely used by mimics for interactions. The relatively fast evolution of target interface residues exclusively used by viral mimics may be related to host discrimination between essential interactions with host-mimicked proteins and deleterious interactions with mimics, providing an escape route for the host through rapid evolution of viral-specific interfaces. However, these interface regions do not display signatures of positive selection that

can serve as evidence of host escape from interactions with viral proteins (Daugherty and Malik, 2012; Duggal and Emerman, 2012; Sironi et al, 2015). Further studies are required to test whether these regions, specifically targeted by viral mimics, evolve in a manner that can weaken these deleterious interactions with viral proteins.

When analyzing the evolution of viral mimics, we observed complex patterns. Across the studied viruses, we did not find consistent trends of evolutionary conservation when comparing viral-mimicking proteins (structural and functional mimicking proteins) and non-mimicking proteins. This lack of consistency can be a result of various technical and biological factors. The fact that the number of viral proteins encoded by each of the viral proteins is significantly smaller than the number of host proteins (despite using some of the largest mammalian viruses) may introduce noise and biases in this analysis. Furthermore, in some of these viruses the range of evolutionary rates is limited due to relatively few available viral orthologs, and due to the need to filter poorly aligned or recombinant regions that would result in erroneous inferences of evolutionary rates. Several functional reasons may also be behind these complex trends, including the fact that viral proteins are often multi-functional; thus by comparing viral mimics with other viral proteins, we do not account for many other functions that can contribute to the evolutionary rates of viral proteins.

In any case, our analysis does not show that viral mimics and their interfaces are enriched with positively selected sites. Furthermore, the fraction of PSGs is lower in mimics than non-mimics. Among the few PSRs detected in mimics, we predicted one mimicking-target protein complex structure, whose PSR is found outside of the interface with the host target. Finally, we note that our analysis is limited to large dsDNA viruses that encode for relatively high numbers of domain and motif mimics, that have sufficient numbers of orthologs, and that have experimental PPI data - information essential for these analyses. Mimics encoded by RNA viruses, which evolve much faster than DNA viruses and can cross between host species more frequently than herpesviruses, may have different evolutionary dynamics with their host.

In general, our results regarding both host and viral proteins involved in mimicry systems suggest that the evolutionary dynamic of these interactions does not usually involve rapid and constant changes, following an evolutionary arms race scenario. Instead, our results point to conservation of the interacting proteins and their interfaces over long evolutionary timescales. This is in agreement with previous findings regarding high degree of conservation of host proteins that interact with viral proteins (Shuler and Hagai, 2022; Dyer et al, 2008), and is in line with suggestions regarding slow evolutionary rates of viral proteins and their relationship to constraints imposed by adaptation to the host (Simmonds et al, 2019).

Finally, our findings regarding conserved interface regions may provide a basis for antiviral treatment that can interfere with interactions mediated through these conserved regions. Notably, our studies point to principles that govern conservation and evolvability of host–virus interaction networks. These are important for predictions of viral emergence and zoonotic transfers, given the dual roles these interactions may play in supporting or preventing host-switches.

# Methods

## Reagents and tools table

| Reagent/resource | Reference or source | Identifier or catalog number |
|---|---|---|
| **Experimental models** | | |
| None | | |
| **Recombinant DNA** | | |
| None | | |
| **Antibodies** | | |
| None | | |
| **Oligonucleotides and other sequence-based reagents** | | |
| None | | |
| **Chemicals, enzymes, and other reagents** | | |
| None | | |
| **Software** | | |
| Guidance | https://github.com/HaimAshk/GUIDANCE | |
| 3SEQ | https://mol.ax/software/3seq/ | |
| PAML | https://github.com/abacus-gene/paml | |
| Selecton | https://academic.oup.com/nar/article/35/suppl_2/W506/2923796 | |
| BLAST | https://blast.ncbi.nlm.nih.gov/Blast.cgi?PROGRAM=blastp&PAGE_TYPE=BlastSearch&LINK_LOC=blasthome | |
| PhyML | https://github.com/stephaneguindon/phyml | |
| AlphaFold2 | https://github.com/google-deepmind/alphafold | |
| AlphaFold-Multimer | https://github.com/google-deepmind/alphafold | |
| CSU | https://oca.weizmann.ac.il/oca-bin/lpccsu | |
| AIUpred | https://aiupred.elte.hu/ | |
| ScanNet | https://bioinfo3d.cs.tau.ac.il/ScanNet/ | |
| ISPRED4 | https://ispred4.biocomp.unibo.it/ispred/default/index | |
| Python v3.12.6 | https://www.python.org/ | |
| Pandas v2.2.2 | https://pandas.pydata.org/ | |
| Numpy v1.26.4 | https://numpy.org/ | |
| Matplotlib v3.9.2 | https://matplotlib.org/ | |
| Seaborn v0.13.2 | https://seaborn.pydata.org/ | |
| Scipy v1.14.1 | https://scipy.org/ | |
| Statsmodels v0.14.2 | https://www.statsmodels.org/stable/index.html | |
| Pymol | https://www.pymol.org/ | |
| **Other** | | |
| None | | |

## Identification of viral domain mimics and host-mimicked proteins

### Host and viral gene assemblies

To identify structural homologs between host and viral proteomes, we first obtained structural predictions of their proteins. As reference proteome sets for structural analyses, we used viral proteomes of the following five dsDNA viruses: herpes simplex virus 1 (*Simplexvirus humanalpha1*, strain 17, UP000009294), Epstein–Barr virus (*Lymphocryptovirus humangamma4*, strain GD1, UP000158635), human cytomegalovirus (*Cytomegalovirus humanbeta5*, Merlin, UP000000938), Kaposi's sarcoma-associated herpesvirus (*Rhadinovirus humangamma8*, GK18, UP000000942), vaccinia virus (*Orthopoxvirus vaccinia*, Western Reserve strain, UP000000344).

### Structural prediction and homology

To predict structures of viral proteins, we ran AlphaFold2 (Tunyasuvunakool et al, 2021) with default parameters and took for each protein the highest-ranking structure (with the highest pLDDT score). For structural comparison with host proteins, we used human protein structural predictions, downloaded from the AlphaFold website (https://alphafold.ebi.ac.uk/).

We then ran FoldSeek (van Kempen et al, 2024) with default parameters. We ran it both using the viral proteome as the query and the human proteome as the target, and vice versa. Taking all pairs of human-viral proteins that were found to be structural homologs of each other, we performed the following stages to exclude spurious matching, matches between simple or small structural domains, or domains that only a portion of them is homologous, making them less likely to be true structural homologs. We required that the fold complexity for the viral protein would be at least 15, as computed by the contact order measure (see below). We required that the ratio in the total length between the two proteins is less than 1.5. We also required that the ratio between the maximum ungapped sequences (where we allow gaps to be up to 5% of the total protein length) between the two proteins will be less than 2.5. We also required that both human and viral proteins would have at least 50aa length. Finally, the median AlphaFold score of the viral protein was required to be at least 70.

Following this filtering, we obtained 7593 pairs of human-viral structural homologs, which are detailed in Dataset EV2. We note that similar approaches employing AlphaFold predictions, structural homology, and subsequent filtering using various criteria were successfully used by previous studies asking various questions regarding structures of viral proteins (Soh et al, 2024; Litvin et al, 2025; Mutz et al, 2023; Nomburg et al, 2024; Kim et al, 2025).

Fold complexity was used as one of the filtering parameters to exclude simple folds that may be found to match spuriously to each other or to parts of other folds. To compute fold complexity, we used a measure, termed "contact order" (Ivankov et al, 2003), that calculates the average sequence distance (in primary sequence) between pairs of interacting residues in the three-dimensional structures. In other words, this measure takes into account the distances between all pairs of interconnected residues, with pairs that interact from distant sequence positions increasing its value and indicating that the fold complexity is higher. Specifically, based on the predicted structures from AlphaFold2, we inferred all

contacting residues using the contacts of structural units (CSU) software (Sobolev et al, 1999) (as previously done (Hagai and Levy, 2008; Hagai et al, 2012)). Using this list of contact residues, and their position in primary sequence, we computed contact order, defined as: $\sum C\,|\,i - j\,|\,/C$, where C is the number of contacts, and |i - j| is the sequence separation between residues i and j, that were found to be in contact. As in previous studies, we ignored contacts between residues that their distance in the primary sequence is below 3 residues (Hagai et al, 2012; Hagai and Levy, 2010).

## Identification of host targets and functionally mimicked proteins

To identify the human proteins mutually targeted by both host-mimicked and viral mimicking proteins, we overlaid experimentally characterized human–virus PPIs with human-human PPIs.

For human–virus interactions, we used the HVIDB database (Yang et al, 2021) that includes 48,643 human–virus interactions originating from different databases and additional curations. For the HCMV analysis, additional human - HCMV interactions were obtained from an additional study (Nobre et al, 2019). For KSHV, we also added interactions curated by BioGrid (Oughtred et al, 2021) and IntAct (Kerrien et al, 2012), since we noticed that many of them were missing in the general host–virus PPI datasets we used. In total, these resulted in 411, 759, 419, 3320, and 1289 interactions between human proteins and VACV, HSV1, EBV, HCMV, and KSHV proteins, respectively.

We obtained the curated list of physical interactions between human proteins (human-human PPIs) from the STRING database (version 11.5) (Szklarczyk et al, 2023). For each gene, we used the interactions for its longest protein isoform, and excluded all data related to other isoforms.

We searched for each host-mimicked and viral-mimicking protein pairs, the set of proteins that are known to experimentally interact with both. These were termed "mutual targets". Host-mimicked proteins known to interact with one such target are termed "functional mimicked", to distinguish them from the superset of "structural mimicked" proteins.

## Identification of viral perfect mimics of host motifs and their host targets

To study evolutionary trends of "perfect mimicry" targets, and to contrast them with the evolution of host targets of imperfect domain mimicry, we used short linear motifs that are embedded in viral proteins and are identical in their sequences and manner of interaction to the host motifs they mimic. To identify triads of viral motif mimics, host-mimicked motifs and mutual targets, that include a motif-binding domain, we employed an approach previously used by us and by others (Garamszegi et al, 2013; Shuler and Hagai, 2022; Hagai et al, 2014; Cagliani et al, 2024; Becerra et al, 2017). In this method, we use experimentally characterized PPIs and identify motifs and their matching binding domains in pairs of interacting proteins.

For this, we first identified motif-matching sequences in disordered regions of viral protein sequences and human protein sequences by searching for linear motifs within the proteins' sequences using regular expression matching (RegEx) of experimentally known motifs, taken from the ELM database (Kumar et al,

2024). To focus only on motifs in disordered regions, we ran AIupred to predict per-residue disorder scores (Erdős and Dosztányi, 2024). Motif-matching sequences were included only if their residues' average AIupred score was above 0.5. To infer which of these motif-matching sequences is likely a functional motif that binds a matching domain, we used existing protein-protein interaction (PPI) databases of human-viral and human-human interactions (for the viral sequences and human sequences analysis, respectively), as described above. Using these PPIs, for each protein that had a motif-matching sequence, we searched for a matching domain in its interactor (e.g., if the viral protein had a sequence that matches an SH3-binding motif, we searched for an SH3 domain in its human interactor). Domain annotations in human proteins were taken from PFAM (Mistry et al, 2021). When such domain-motif matches were found, the motifs were considered as "inferred functional motifs". If the same human protein was found to interact through domain-motif interactions with inferred functional motifs from both viral and human interactors, then this human protein was considered a target of motif mimicry. The set of human proteins targeted by motif mimicry is denoted as "motif targets", as opposed to "domain targets", which are used throughout the manuscript.

## Evolutionary analysis of host proteins

### Host gene orthology assignment

To infer selection patterns across host proteins, we analyzed human genes and their non-human primate orthologs. To compare amino acid substitutions across the primate clade, we used the human transcriptome as a reference and a set of nine non-human primate transcriptomes: chimpanzee, northern white-cheeked gibbon, common marmoset, rhesus macaque, olive baboon, western lowland gorilla, Sumatran orangutan, crab-eating macaque, and gray mouse lemur. Transcriptomes and orthology relationships were collected using ENSEMBL (version 110)(Martin et al, 2023). The chosen set of non-human primates was based on their genome annotation quality, having a relatively large number of annotated genes and a high N50 value (a measure that describes the "completeness" of a genome assembly). We used a set of 10 primates to have a sufficiently large number of species to reflect the diversity of the primate clade and to enable a rigorous analysis of protein evolution over this timescale. We chose not to include additional species, since the addition of each species reduces the number of genes that have one-to-one orthologs. The subsequent evolutionary analysis was performed only with genes that have one-to-one orthologs across all 10 species, a total of 10,544 genes. For every human gene, the transcript with the lowest TSL score (Transcript Support Level) from ENSEMBL was selected as the representative transcript. For the other nine primates, the best-matching transcript to the human ortholog according to BLAST (Altschul et al, 1990) was selected.

### Sequence alignment and tree reconstruction

Multiple Sequence Alignments (MSAs) were created for the ortholog sequences using GUIDANCE (version 2.02) (Sela et al, 2015), with masking of unreliably aligned positions in the MSA and using default parameters of 100 bootstrap iterations, sequence default cutoffs of <60% and columns cutoffs of <93% (with the parameters: --program GUIDANCE2 --seqType codon

--msaProgram MAFFT"). 3SEQ program was used to identify recombination points in the sequences (Lam et al, 2018). 3SEQ was used in full-run mode, with a default pre-computed $700 \times 700 \times 700$ $P$ value table. (3seq -f "input_msa" -id "result_file"). Gene regions identified as products of recombination events were masked to minimize recombination effects on positive selection results. Masking was performed by changing these residues of the MSA to 'N's.

A phylogenetic tree of primates was built based on a pre-computed vertebrate tree from ENSEMBL (https://www.ensembl.org/info/about/speciestree.html) (Herrero et al, 2016).

### Evolutionary rate inference

To compute dN/dS values, we used Selecton (version 2.4) (Stern et al, 2007) that performs accurate Bayesian rate estimations with prior Bayesian distribution of beta + w (Selecton -i [input file] -q [query sequence] -u [tree]). We obtained dN/dS values for each residue, using the M8 evolutionary model, in 10,531 one-to-one orthologs in all 10 species (in 13 genes, the run failed to complete due to having sequences that were too long). Next, we filtered out genes whose MSAs were overly masked, defined as those in which the masked region exceeds 50% of the total MSA length, based on either all primate sequences in the MSA or only the human sequence. After filtering, 9,178 genes remained, and were included in the analysis.

### Positive selection analysis

Of the 10,544 human genes with MSAs, 10,540 were analyzed by PAML (4 genes have failed during the run). Next, we filtered out genes whose MSAs were over-masked. Over-masked MSAs were defined as those in which the masked or gapped region exceeds 50% of the total MSA length, based on either all sequences in the MSA or only the reference human sequence. After filtering, 8664 genes were included in the analysis. To detect positive selection, the codon-based *codeml* program implemented in the PAML suite was applied (Yang, 2007). Using F3 $\times$ 4 codon frequencies model (codon frequencies estimated from the nucleotide frequencies in the data at each codon site) (Yang, 2007), M8 model (positive selection model) that allows a class of sites to evolve with dN/dS > 1 was compared to M8a model (neutral model) that limits dN/dS $\leq$ 1 (*codeml* "input_ctl_file"). To assess statistical significance, twice the difference of the likelihood ($\Delta$lnL) for the models (M8a versus M8) is compared to a $\chi^2$ distribution (1 degree of freedom). To obtain robust results, we determined positive selection at the gene level using four q-value thresholds (q-value < 0.05, 0.01, 0.001, 0.0001).

## Evolutionary analysis of viral proteins

### Viral gene orthology assignment

Genome sequences for each viral genus (*Orthopoxvirus*, *Simplexvirus*, *Cytomegalovirus*, and *Rhadinovirus*) were retrieved from the National Center for Biotechnology Information database (NCBI, http://www.ncbi.nlm.nih.gov/). Only complete (or almost complete) genome sequences were included (see Table EV2 for a complete list of viral species and strains used). In the case of VACV, we aligned VACV protein residues (Western Reserve

sequence) with the MSA (that included horsepox virus as the VACV sequence (Molteni et al, 2023)) to match between structure-based analysis and evolutionary analysis.

For each viral genus, we used Progressive Mauve (v.2.3.1) (Darling et al, 2010) to identify one-to-one orthologous genes. Mauve identifies and aligns genomic regions of local collinearity (locally collinear blocks, LCBs). Each LCB is a region of sequence homology shared by two or more of the genomes being aligned. Mauve was run using default parameters. Orthology was inferred according to Mauve attribution and validated by genome annotation (if available).

### Sequence alignment and tree reconstruction

Once orthologous sequences were identified, gene alignments were generated using the GUIDANCE2 suite (Sela et al, 2015) and recombination analysis was performed using the 3SEQ Recombination Detection Algorithm (Lam et al, 2018), as described above. We considered only recombining segments with a length >100 nt and a $P$ value < 0.01, that were masked for subsequent analyses. Next, we filtered out genes whose MSAs were overly masked, as described in the host evolutionary analysis section, leaving 122, 72, 79, and 71 genes for orthopoxviruses, simplexviruses, cytomegaloviruses, and rhadinoviruses, respectively.

Phylogenetic trees were reconstructed by maximum likelihood using phyML (Guindon et al, 2010). We set a General Time Reversible (GTR) model plus gamma-distributed rates and four substitution rate categories, a fixed proportion of invariable sites, and a BioNJ starting tree.

### Evolutionary rate inference

To compute dN/dS values, we used Selecton (version 2.4) (Stern et al, 2007), as described in the host evolutionary analysis. We obtained dN/dS values for each residue, using the M8 evolutionary model, in all one-to-one orthologs in all species, for each virus- 119, 72, 79, and 71 genes for orthopoxviruses, simplexviruses, cytomegaloviruses, and rhadinoviruses, respectively (in 3 orthopoxvirus genes the run failed completion).

### Positive selection analysis

To detect positively selected genes and residues, we performed a similar analysis as described in the host evolutionary analysis section. Briefly, we used the codon-based *codeml* program implemented in the PAML suite for site-models (Yang, 2007).

We used branch-site analysis data available for each viral genus among herpesviruses (Mozzi et al, 2022, 2020, 2025) to explore adaptive evolution in lineages infecting hominin species.

## Analysis of human gene and protein characteristics

### Gene expression analysis

We downloaded gene expression data from the Human Protein Atlas (Uhlén et al, 2015). Gene expression levels were obtained from 43 tissues of healthy human adults. We calculated the mean expression per gene across tissues (using TPMs—transcript per million measure). We further computed the number of tissues expressing each gene, defined as all tissues with a gene expression level of 3 TPMs or higher.

### Gene essentiality analysis

We used a dataset of 1093 essential genes, assembled by Bartha et al (Bartha et al, 2018). In this work, genes were defined as essential using several metrics, including data from in vitro and in vivo studies. We considered a gene to be essential if it was found to be essential in at least one of the three main screens used in this work.

### Functional enrichment analysis

We used gProfiler (Raudvere et al, 2019) to test for functional enrichment of host targets and mimicked proteins against all human proteins. For each of the significantly enriched terms in the biological processes category, we then tested whether targets or mimicked proteins are more conserved than the rest of the human proteins belonging to this term, using Mann–Whitney test and correcting using FDR (Dataset EV5). From these lists, non-redundant term lists are shown with their dN/dS distributions in Fig. 2F,G (taking the most significant term from each group of related GO terms).

## Protein complex prediction and analysis

### Prediction of protein complex structures

To predict protein complex structures between host-mimicked and targets and viral-mimicking and targets, we employed AlphaFold-Multimer (Evans et al, 2021) for each such protein pair (576 host-mimicked–target and 116 viral-mimicking–target). From these predictions, we took the highest-ranking predicted structure. Protein complexes whose prediction scores were sufficiently high ($0.8 \times \text{ipTM} + 0.2 \times \text{pTM} > 0.7$) were then used in subsequent analyses, where their interface residues were inferred based on the structures. These included 59 host-mimicked–target complexes and 12 viral-mimicking–target complexes.

### Structural analysis of protein complexes and interface inference

In order to define residues as belonging to the surface, core or interface of each of the proteins within each complex using structural analysis based on the CSU software (Sobolev et al, 1999). CSU solvation measurements were used to calculate the solvent accessible surface area (ASA). ASA was defined as the ratio of the solvent accessible surface for a given residue within the structured protein versus in the free-state of that residue. Residues were classified as core residues for ASA < 0.15, and as surface residues for ASA > 0.15, as previously done (Tóth-Petróczy and Tawfik 2011; Shuler and Hagai, 2022). CSU also identifies residues that come into contact with each other. This was used to infer interface residues between host-mimicked and target proteins, and between viral mimicking and target proteins.

In order to identify interface regions of host-mimicked proteins or of target proteins with other proteins, unrelated to mimicry, we used the ScanNet program (Tubiana et al, 2022). ScanNet is a geometric deep learning model for prediction protein-protein interaction sites. We used the following parameters to define residues as belonging to interface regions—a minimal score of 0.7 across all residues for Main Fig. 3C, and a minimal score of 0.7 or 0.5, with all residues or only with ordered residues, for Fig. EV2. These interface regions are defined as "general-interface" regions in the relevant sections. We further used the ISPRED4 program (Savojardo et al, 2017) as an additional method to infer interface regions. This method uses a different model than ScanNet, based on the combination of support vector machines (SVMs) and grammar-restrained conditional random fields (GRHCRF). The program was run using default parameters (RSA > = 0.20), and a minimal interface probability of 0.7 was required for a residue to be defined as interface.

## Analysis of viral gene and protein characteristics

We partitioned viral proteins based on several characteristics that were previously shown to be associated with their evolutionary rates, and based on whether the proteins are viral mimicking or not. These included core versus non-core proteins or evolutionary age in all four viruses, taken from (Fuchs et al, 2025; Davison et al, 2009; Molteni et al, 2023), and characteristics that were studied in the three herpesvirus proteomes—protein fold complexity, fraction of disordered regions and time of expression (all values were taken from the same study (Fuchs et al, 2025)). In each of these cases, mimicking and non-mimicking proteins were compared to test whether they differ in occurrence across different classes (temporal expression and core vs accessory partition) or in values (fold complexity and disorder fraction).

## Statistical analysis

Statistical analyses (Mann–Whitney test, Fisher's exact test, Spearman's rank correlation and FDR-correction based on Benjamini-Hochberg procedure (Benjamini and Hochberg, 1995) were performed using the SciPy package in Python (version 3.12). Data in boxplots represent the distribution with lines for median, first quartile and third quartile and whisker lines extending to the furthest value within 1.5 of the interquartile range. Violin plots show the kernel probability density of the data. Plots were created using matplotlib, seaborn, and Plotly packages.

# Data availability

MS, MSA and tree files, and PDB files of AlphaFold predicted structures are available on Zenodo: https://zenodo.org/records/17936739. Code generated during this study and command lines of relevant programs are available at: https://github.com/HagaiLab/domain_mimicry. Input files required for running these scripts can be found on the following Zenodo link: https://zenodo.org/records/17936997.

The source data of this paper are collected in the following database record: biostudies:S-SCDT-10_1038-S44320-026-00200-1.

# Peer review information

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

## Acknowledgements

We would like to thank Ron Geller for helpful comments on the manuscript. This work was supported by the Italian Ministry of Health ("Ricerca Corrente" to DF and AM), by the Israel Science Foundation (ISF, grant No. 435/20 to TH), and by the Joint Canada-Israel Research Program (ISF, grant No. 2930/23 to TH). The work was also funded by the European Union (ERC, EvoViralMimicry, 101171091 to TH). Views and opinions expressed are however those of the authors only and do not necessarily reflect those of the European Union or the European Research Council. Neither the European Union nor the granting authority can be held responsible for them.

## Author contributions

**Rotem Fuchs**: Data curation; Formal analysis; Investigation; Visualization; Writing—original draft. **Ofir Schor**: Data curation; Formal analysis; Investigation; Writing—original draft. **Bar Naim**: Data curation; Formal analysis; Investigation. **Dafna Tussia-Cohen**: Data curation; Formal analysis; Investigation. **Alessandra Mozzi**: Data curation; Formal analysis; Investigation. **Diego Forni**: Data curation; Formal analysis; Investigation. **Sivan Friedman**: Data curation. **Zohar Haggai**: Data curation. **Manuela Sironi**: Supervision; Investigation; Methodology; Writing—original draft; Writing—review and editing. **Tzachi Hagai**: Conceptualization; Formal analysis; Supervision; Funding acquisition; Investigation; Methodology; Writing—original draft; Writing—review and editing.

Source data underlying figure panels in this paper may have individual authorship assigned. Where available, figure panel/source data authorship is listed in the following database record: biostudies:S-SCDT-10_1038-S44320-026-00200-1.

## Disclosure and competing interests statement

The authors declare no competing interests.

# Expanded View Figures

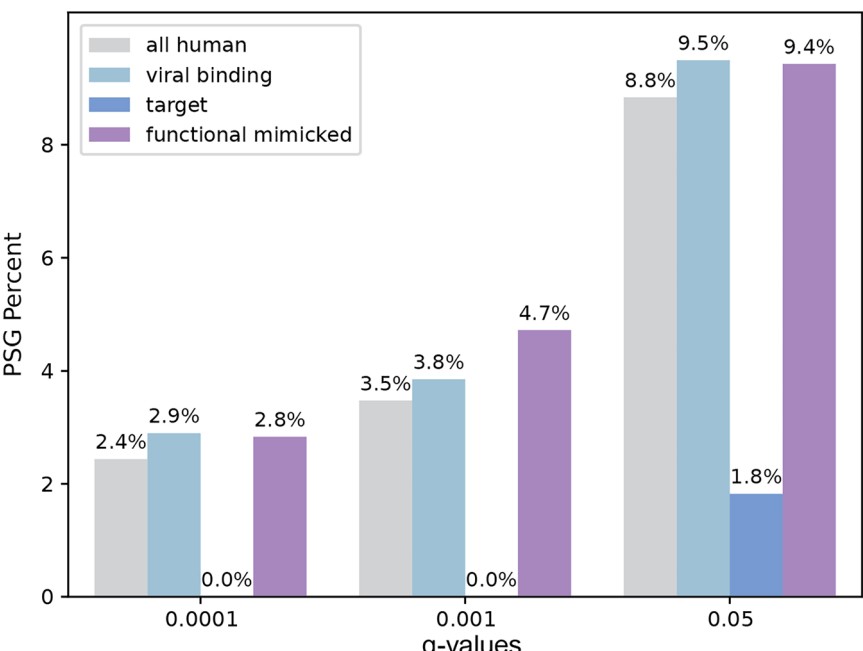

**Figure EV1.  Bar plots showing percentage of genes within each group, with signatures of positive selection (PSGs), as follows.**

All human proteins (9474 proteins), human proteins experimentally known to interact with at least one viral protein from the set of the five dsDNA viruses used in this study (viral binding, 3599 proteins), human proteins known to interact with viral mimics and host-mimicked proteins (mutual targets, 59 proteins), human proteins that have at least one structural homolog in the five viral proteomes used in this study (structural mimicked, 605 proteins), a subset of structural mimicked that are also known to interact with the same target as viral mimicking proteins (functional mimicked, 111 proteins). The evolutionary rate is relative across all proteins residues (5,751,638 residues in total) and is computed based on substitutions across a set of 10 one-to-one orthologs in primates, using the ratio between the number of non-synonymous and synonymous substitutions (dN/dS values, left). PSGs were identified based on a likelihood ratio test between two models (M8 versus M8A), and based on a statistical significance threshold. Three different thresholds are shown (from left to right, FDR-corrected *P* values of 0.0001, 0.001 and 0.05). Percentage of PSGs with FDR-corrected *P* value of 0.01 is shown in Fig. 2B. Statistical enrichment (or depletion) was computed for each gene subset with respect to the group of all human genes using Fisher's exact test and corrected by FDR. None of the comparisons resulted in a significant *P* value (i.e., all *P* values > 0.05). The total numbers of PSGs in each of the categories - all human, viral binding, target and functional mimicked, are in: FDR-corrected *P* values of 0.0001: 211,103,0,3; and FDR-corrected *P* values of 0.001: 301,137,0,5; and FDR-corrected *P* values of 0.05: 766,338,1,10, respectively. /ppImages were created in BioRender. (2025) https://BioRender.com/1m65sw7.

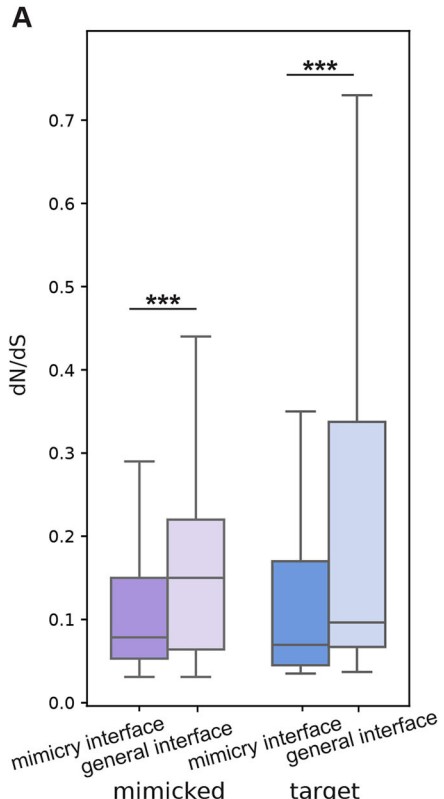

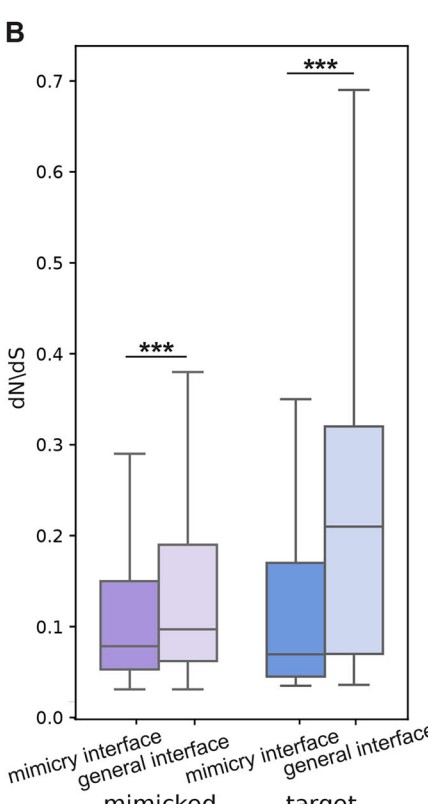

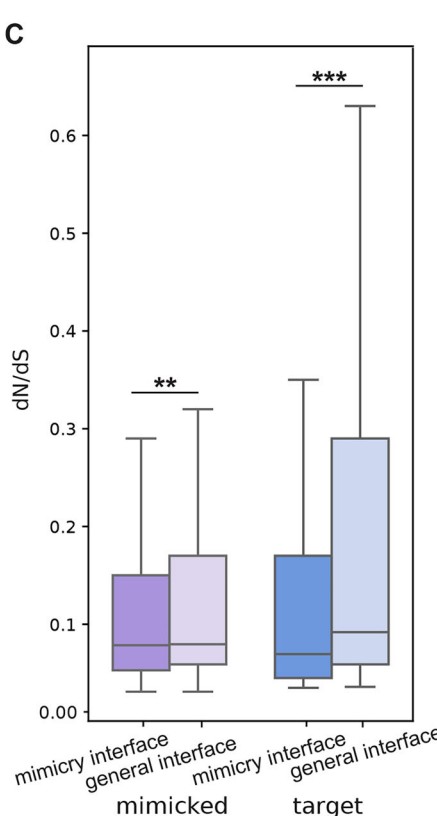

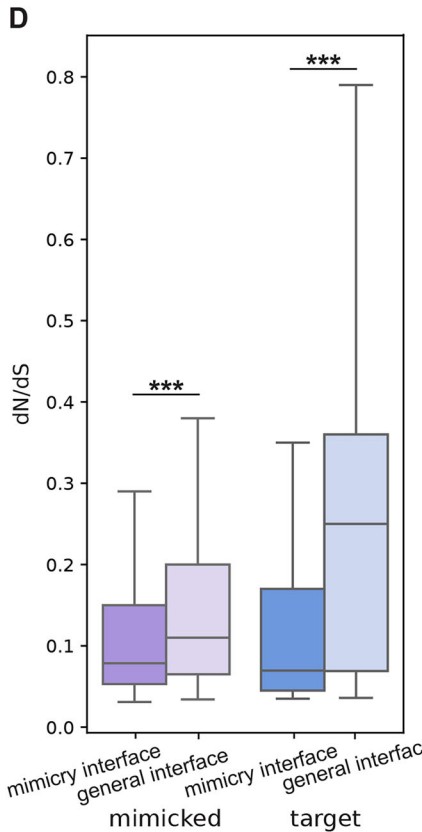

◀  **Figure EV2.  Comparison of evolutionary rates between mimicry interface and general interface residues, in mimicked and target proteins.**

As in Fig. 3C, with the mimicry-interface residues predicted based on the complexes between host-mimicked and target proteins using the CSU method, and the general-interface residues predicted by ScanNet, but with different set of parameters: (A) ScanNet score of 0.7 and above is used to determine interface residues, only residues predicted to be ordered regions are considered (unlike in Fig. 3C, where both disordered and ordered residues were considered). *P* values for comparisons of mimicked general interface vs. mimicked mimicry interface, and target general interface vs. target mimicry interface are 1.44e-06 and 2.33e-10, respectively. *N* of residues in mimicked–general interface, mimicry interface and target–general interface, mimicry interface are 145,642,218,762 residues, respectively. (B) ScanNet score of 0.5 and above is used to determine interface residues, all residues (both ordered and disordered) are considered. *P* values for comparisons of mimicked general interface vs. mimicked mimicry interface, and target general interface vs. target mimicry interface are 2.63e-07 and 1.70e-42, respectively. *N* of residues in mimicked–general interface, mimicry interface and target–general interface, mimicry interface are 1014,642,1147,762 residues, respectively. (C) ScanNet score of 0.5 and above is used to determine interface residues, only residues predicted to be ordered regions are considered. Groups were compared using Mann–Whitney test and corrected by FDR. *P* values for comparisons of mimicked general interface vs. mimicked mimicry interface, and target general interface vs. target mimicry interface are 1.29e-03 and 5.70e-09, respectively. *N* of residues in mimicked general interface, mimicry interface and target general interface, mimicry interface are 478,642,437,762 residues, respectively. (D) ISPRED4 probability of 0.7 and above is used to determine interface residues (all residues were considered). $^{***}P < 0.001$, $^{**}P < 0.01$, $^{*}P < 0.05$. Boxplots in (A–D) represent the median, first quartile and third quartile with lines extending to the furthest value within 1.5 of the interquartile range (IQR). *P* values for comparisons of mimicked general interface vs. mimicked mimicry interface, and target general interface vs. target mimicry interface are 2.58e-14 and 4.90e-55, respectively. *N* of residues in mimicked–general interface, mimicry interface and target–general interface, mimicry interface are 1335,642,1525,762 residues, respectively.

                                      

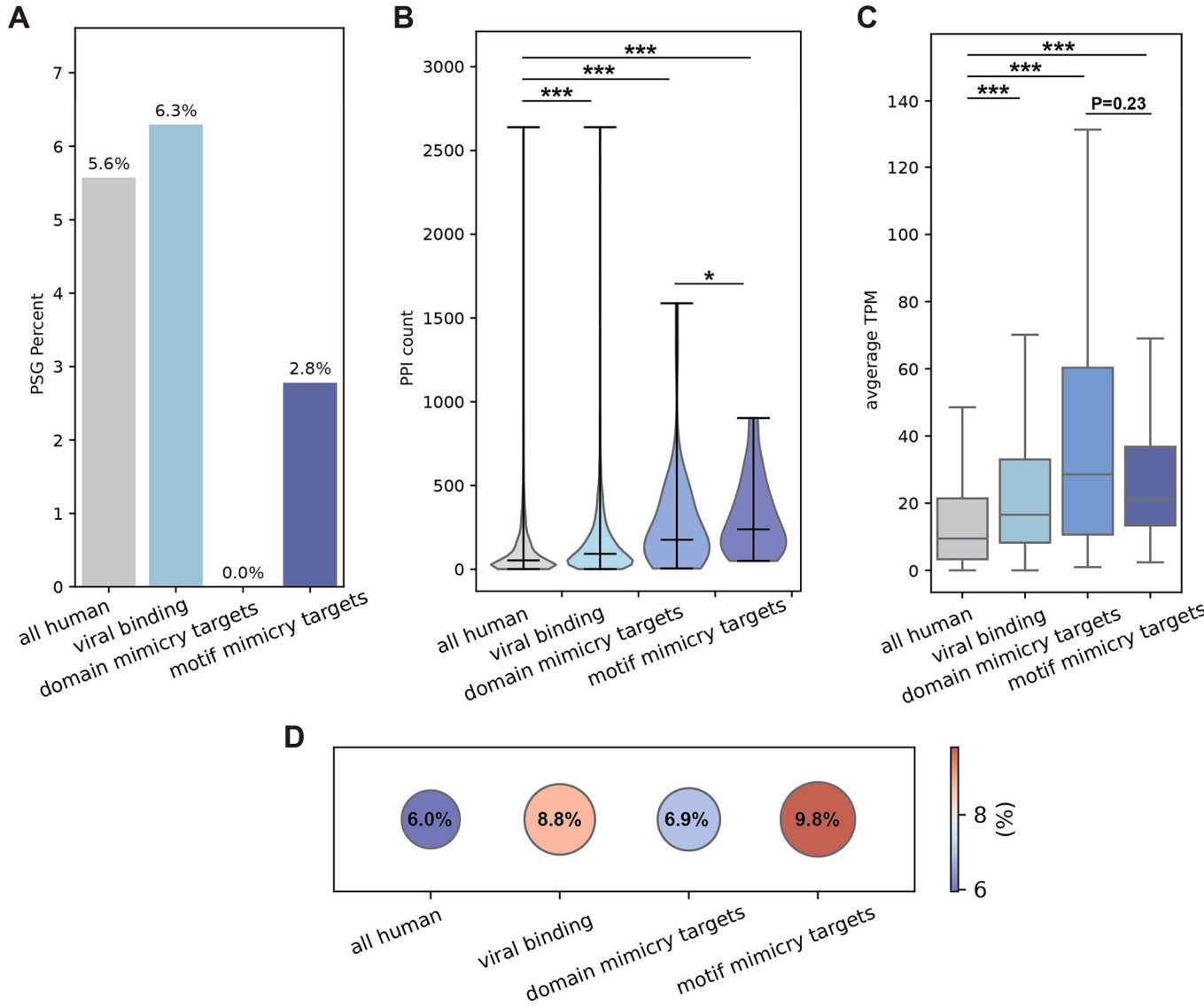

**Figure EV3. Comparison of evolutionary, functional and cellular characteristics between different groups of human proteins, including targets of motif mimicry.**

(A) Bar plots showing percentage of genes within each group, with signatures of positive selection (PSGs), as follows: all human proteins (9173 proteins), human proteins experimentally known to interact with at least one viral protein from the set of the five dsDNA viruses used in this study (viral binding, 3489 proteins), human proteins known to interact with viral domain mimics and host-mimicked domains (domain-mimicry targets, 58 proteins), human proteins known to interact with viral motif mimics and host-mimicked motifs (motif-mimicry targets, 49 proteins). Statistical enrichment (or depletion) was computed for each gene subset with respect to the group of all human genes using Fisher's exact test and corrected by FDR. None of the comparisons resulted in a significant *P* value, except for viral-binding versus all human proteome. (B) Violin plots showing the number of within-host PPIs for the sets of proteins defined in (A) (each filled area extends to represent the entire data range). *P* values for comparisons of all vs. viral binding, domain mimicry targets, motif mimicry targets and domain mimicry targets vs. motif mimicry targets are 1.67e-86, 8.733e-12, 2.775e-17, 3.99e-02, respectively. (C) Boxplots showing the distributions of average gene-expression levels across healthy adult human tissues for each gene group described in (A). *P* values for comparisons of all vs. viral binding, domain mimicry targets, motif mimicry targets, and domain mimicry targets vs. motif mimicry targets are 4.4e-125, 6.48e-009, 5.934e-008, 2.266e-001, respectively. (D) Dotplots showing the fraction of essential genes within each set of genes defined in A. None of the comparisons resulted in a significant, except for viral-binding versus all human proteome (*P* value = $5.4 \times 10^{-8}$). In all panels: ***$P < 0.001$, **$P < 0.01$, *$P < 0.05$. Boxplots in (C) represent the median, first quartile and third quartile with lines extending to the furthest value within 1.5 of the interquartile range (IQR).

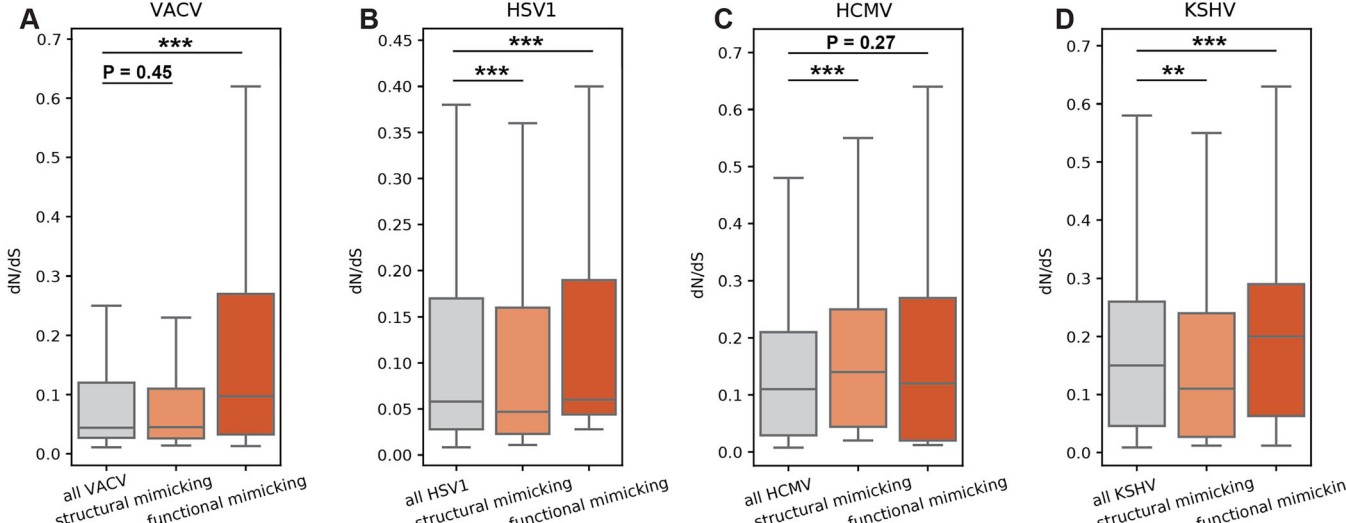

**Figure EV4.  Boxplots showing the distribution of evolutionary rates of different viral protein sets, comparing rates between all viral proteins, structural-mimicking, and functional-mimicking proteins.**

(**A**) VACV (representing orthopoxviruses; all proteins—113, structural mimicking—33, functional mimicking—3), *P* values for comparisons of all VACV vs. structural mimicking, functional mimicking are 0.45, 8.618e-41, respectively. *N* of residues for all VACV, structural mimicking, functional mimicking are 33320, 10093, 1052, respectively. (**B**) HSV1 (simplexviruses—all proteins—69, structural mimicking—5, functional mimicking—3), *P* values for comparisons of all HSV1 vs. structural mimicking, functional mimicking are 7.109e-25, 2.89e-21, respectively. *N* of residues for all HSV, structural mimicking, functional mimicking are 36219, 3214, 1221 residues, respectively. (**C**) HCMV (cytomegaloviruses—all proteins—78, structural mimicking—4, functional mimicking— 2), *P* values for comparisons of all HCMV vs. structural mimicking, functional mimicking are 2.509e-34, 2.721e-01, respectively. N of residues for all HCMV, structural mimicking, functional mimicking are 38217, 2192, 750 residues, respectively. (**D**) KSHV (rhadinoviruses—all proteins—98, structural mimicking—5, functional mimicking—8). *P* values for comparisons of all KSHV vs. structural mimicking, functional mimicking are 2.034e-12, 9.518e-26, respectively. *N* of residues for all KSHV, structural mimicking, functional mimicking are 32256, 1849, 2253 residues, respectively. Evolutionary rates across each lineage were computed using Selecton and MSAs based on one-to-one viral orthologs, as described in Methods. In these plots, viral proteins that have significant structural domain homology to at least one human protein were considered as mimicking, and are here partitioned into structural- and functional-mimicking, where they are mutually exclusive and the partition is based on whether or not the viral-mimicking protein has a shared interactor with the host-mimicked protein. Groups were compared using Mann–Whitney test and corrected by FDR. ***P* < 0.001, **P* < 0.01, *P* < 0.05. Boxplots in (**A–D**) represent the median, first quartile and third quartile with lines extending to the furthest value within 1.5 of the interquartile range (IQR).

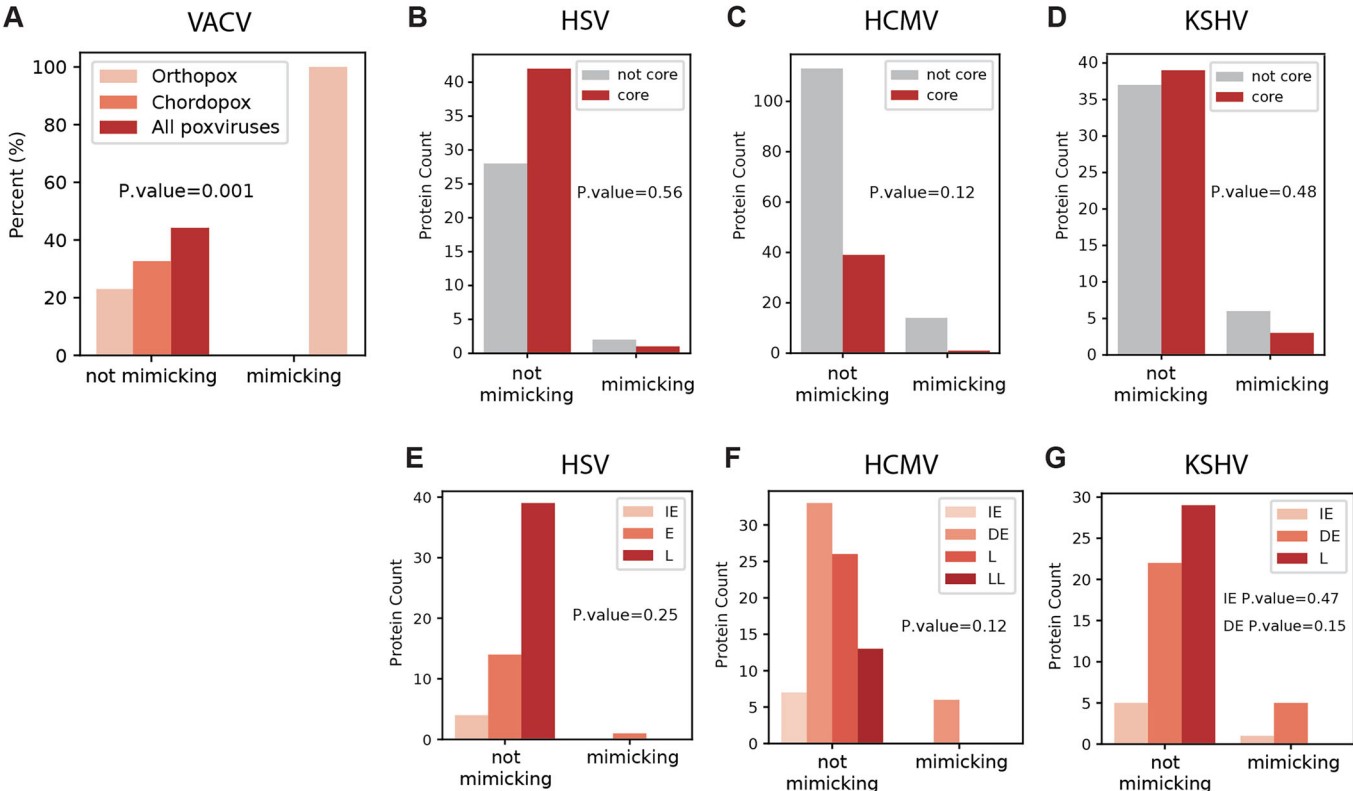

**Figure EV5.  Occurrence of mimicking and non-mimicking proteins in various functional groups.**

(**A**) Occurrence of VACV proteins, partitioned based on functional versus non-functional mimicking proteins in different evolutionary ages. (**B–D**) Occurrence of herpesvirus proteins, partitioned based on functional versus non-functional mimicking proteins, and partitioned based on core vs non-core proteins. (**E–G**) Occurrence of herpesvirus proteins, partitioned based on functional versus non-functional mimicking proteins, and partitioned based on temporal gene expression.

