## [Peer Review File · Molecular Systems Biology]

The evolutionary dynamics between viral mimics and host proteins

Rotem Fuchs, Ofir Schor, Bar Naim, Dafna Tussia-Cohen, Alessandra Mozzi, Diego Forni, Sivan Friedman, Zohar Haggai, Manuela Sironi, and Tzachi Hagai

Corresponding author(s): Tzachi Hagai (tzachiha@tauex.tau.ac.il)

Review Timeline:

Submission Date:	23rd Sep 25
Editorial Decision:	5th Nov 25
Revision Received:	16th Dec 25
Editorial Decision:	30th Jan 26
Revision Received:	7th Feb 26
Accepted:	10th Feb 26

Editor: Jingyi Hou

Transaction Report:

5th Nov 2025

Manuscript Number: MSB-2025-13373

Title: The evolutionary dynamics between viral mimics and host proteins

Author: Rotem Fuchs

Ofir Schor

Bar Naim

Dafna Tussia-Cohen

Alessandra Mozzi

Diego Forni

Sivan Friedman

Zohar Haggai

Manuela Sironi

Tzachi Hagai

Dear Prof Hagai,

Thank you for submitting your work to Molecular Systems Biology. We have now heard back from the three reviewers who agreed to evaluate your manuscript. As you will see from the comments below that they find the study interesting and relevant. They raise, however, several important points, which should be convincingly addressed in a revision of this work.

I think that the recommendations of the reviewers are rather straightforward so there is no need to repeat the points listed below. All issues raised by the reviewers need to be satisfactorily addressed. As you may already know, our editorial policy allows in principle a single round of major revision so it is essential to provide responses to the reviewers' comments that are as complete as possible. Please feel free to contact me in case you would like to discuss in further detail any of the issues raised by the reviewers.

On a more editorial level, we would ask you to address the following issues:

- Please provide a .docx formatted version of the manuscript text (including legends for main figures, EV figures and tables). Please make sure that the changes are highlighted to be clearly visible.
- Please provide individual production quality figure files as .eps, .tif, .jpg (one file per figure).
- Please provide a .docx formatted letter INCLUDING the reviewers' reports and your detailed point-by-point responses to their comments. As part of the EMBO Press transparent editorial process, the point-by-point response is part of the Review Process File (RPF), which will be published alongside your paper.
- Please note that all corresponding authors are required to supply an ORCID ID for their name upon submission of a revised manuscript.
- We replaced Supplementary Information with Expanded View (EV) Figures and Tables that are collapsible/expandable online (see examples in <http://msb.embopress.org/content/11/6/812>). A maximum of 5 EV Figures can be typeset. EV Figures should be cited as 'Figure EV1, Figure EV2' etc... in the text and their respective legends should be included in the main text after the legends of regular figures.

Additional Tables/Datasets should be labeled and referred to as Table EV1, Dataset EV1, etc. Legends have to be provided in a separate tab in case of .xls files. Alternatively, the legend can be supplied as a separate text file (README) and zipped together with the Table/Dataset file.

For the figures and tables that you do NOT wish to display as Expanded View figures, they should be bundled together with their legends in a single PDF file called *Appendix*, which should start with a short Table of Content. Each legend should be below the corresponding Figure/Table in the Appendix. Appendix figures and tables should be referred to in the main text as: "Appendix Figure S1, Appendix Figure S2, Appendix Table S1" etc. See detailed instructions regarding expanded view here: <https://www.embopress.org/page/journal/17444292/authorguide#expandedview>.

- Before submitting your revision, primary datasets (and computer code, where appropriate) produced in this study need to be deposited in an appropriate public database (see <http://msb.embopress.org/authorguide-dataavailability> <https://www.embopress.org/page/journal/17444292/authorguide#dataavailability>).

The accession numbers and database should be listed in a formal "Data Availability" section (placed after Materials & Method) that follows the model below (see also <https://www.embopress.org/page/journal/17444292/authorguide#dataavailability>). Please

note that the Data Availability Section is restricted to new primary data that are part of this study.

Data availability

- RNA-Seq data: Gene Expression Omnibus GSE46843 (<https://www.ncbi.nlm.nih.gov/geo/query/acc.cgi?acc=GSE46843>)

- [data type]: [name of the resource] [accession number/identifier/doi] ([URL or identifiers.org/DATABASE:ACCESSION])

Additional information on source data and instruction on how to label the files are available

- Our journal encourages inclusion of *data citations in the reference list* to directly cite datasets that were re-used and obtained from public databases. Data citations in the article text are distinct from normal bibliographical citations and should directly link to the database records from which the data can be accessed. In the main text, data citations are formatted as follows: "Data ref: Smith et al, 2001". In the Reference list, data citations must be labeled with "[DATASET]". A data reference must provide the database name, accession number/identifiers and a resolvable link to the landing page from which the data can be accessed at the end of the reference. Further instructions are available at .

- We updated our journal's competing interests policy in January 2022 and request authors to consider both actual and perceived competing interests. Please review the policy <https://www.embopress.org/competing-interests> and update your competing interests if necessary.

Please use the heading "Disclosure statement and competing interests".

- All Materials and Methods need to be described in the main text using our 'Structured Methods' format. According to this format, the Methods section includes a Reagents and Tools Table (listing key reagents, experimental models, software and relevant equipment and including their sources and relevant identifiers) followed by a Methods and Protocols section describing the methods, ideally using a step-by-step protocol format. The aim is to facilitate adoption of the methodologies across labs.

Please download and fill our Reagents and Tools Table template (.docx), which you can find in our author guidelines:

<https://www.embopress.org/page/journal/17444292/authorguide#structuredmethods>.

-Regarding data quantification:

Please ensure to specify the name of the statistical test used to generate error bars and P values, the number (n) of independent experiments (please specify technical or biological replicates) underlying each data point and the test used to calculate p-values in each figure legend. Discussion of statistical methodology can be reported in the materials and methods section, but figure legends should contain a basic description of n, P and the test applied.

Graphs must include a description of the bars and the error bars (s.d., s.e.m.).

- Please provide a "standfirst text" summarizing the study in one or two sentences (approximately 250 characters, including space), three to four "bullet points" highlighting the main findings and a "synopsis image" (550px width and 400-600 px height, PNG format) to highlight the paper on our homepage.

Here are a couple of examples:

<https://www.embopress.org/doi/10.15252/msb.20199356>

<https://www.embopress.org/doi/10.15252/msb.20209475>

<https://www.embopress.org/doi/10.15252/msb.209495>

When you resubmit your manuscript, please download our CHECKLIST (<https://www.embopress.org/pb-assets/embo-site/EMBO%20Press%20Author%20Checklist-1642513524327.xlsx>) and include the completed form in your submission.

Please note that the Author Checklist will be published alongside the paper as part of the transparent process (<https://www.embopress.org/page/journal/17444292/authorguide#transparentprocess>).

If you feel you can satisfactorily deal with these points and those listed by the referees, you may wish to submit a revised version of your manuscript. Please attach a covering letter giving details of the way in which you have handled each of the points raised by the referees. A revised manuscript will be once again subject to review and you probably understand that we can give you no guarantee at this stage that the eventual outcome will be favorable.

I look forward to receiving your revised manuscript soon.

Sincerely,
Jingyi

Jingyi Hou, PhD
Senior Editor
Molecular Systems Biology

We realize that it is difficult to revise to a specific deadline. In the interest of protecting the conceptual advance provided by the work, we recommend a revision within 3 months (3rd Feb 2026). Please discuss the revision progress ahead of this time with the editor if you require more time to complete the revisions.

*** PLEASE NOTE *** As part of the EMBO Press transparent editorial process initiative (see our Editorial at <https://dx.doi.org/10.1038/msb.2010.72>), Molecular Systems Biology publishes online a Review Process File with each accepted manuscripts. This file will be published in conjunction with your paper and will include the anonymous referee reports, your point-by-point response and all pertinent correspondence relating to the manuscript. If you do NOT want this File to be published, please inform the editorial office at contact@molsystbiol.org within 14 days upon receipt of the present letter.

Reviewer #1:

Fuchs et al. perform evolutionary analyses on a large number of viral mimics and the host proteins they interact with and mimic structurally. Starting with the proteomes of five large dsDNA viruses, they use structural modeling to predict a set of viral proteins that mimic host proteins, and go on to predict the host proteins that these viral mimics likely interact with. This defines a set of host genes involved in viral mimicry that they perform evolutionary analyses on to determine how these evolve across primates compared to other genes that are either known to interact with viruses or not. They further define a set of 59 predicted host-virus complexes that they model structurally to analyze the evolution of residues at, or away from, the predicted virus-host interaction interfaces. Finally, they show that viral mimics are relatively well conserved within viral genomes, especially in the interface residues that interact with host proteins.

Overall, this is a well performed set of analyses and represents a large dataset of evolutionary data on viral mimics and the host proteins they mimic or interact with. While several of the conclusions have been proposed previously for individual virus mimic examples, the size of the dataset lends credence to the generalizable nature of the conclusions. Importantly, the authors also make several efforts to caveat their conclusions so as to not over-interpret their data, although one important caveat that is not raised is the reliance on DNA viruses, and mostly herpesviruses, exclusively. Additional comments to improve the readability or utility of the paper are described below.

Major concerns:

- 1) Several points in the manuscript could benefit from being more explicit about how analyses were performed, either in the results of the methods sections. For instance, on page 5, the authors describe 540 viral proteins that went into their analysis. Where did this list come from? Can the authors include this list in a supplemental table? Similarly, on page 10, they describe using AlphaFold-multimer and getting a list of 59 protein complexes, but it is unclear how many were used as starting points. In general, it would help readability to be as transparent as possible with what the starting and ending lists are for all of these analyses.
- 2) The authors describe their AlphaFold confidence threshold as ($ipTM+pTM>0.7$). Such a threshold would be very low, as each of those values could be <0.4 . If that is the case, the threshold needs to be higher. However, I am assuming the authors mean that $ipTM<0.7$ and $pTM<0.7$. If so, please clarify.
- 3) The ScanNet program, and exactly what it is predicting, needs to be described in more detail given how it is being used to "predicts binding sites on protein surfaces". Is there an alternate way to predict binding sites on proteins? I appreciate that the authors set several different thresholds for their ScanNet outputs, but it appears to be largely a black box in terms of what is exactly being predicted, which is insufficient to then draw conclusions from.
- 4) As mentioned in the summary, the one major caveat to this study is that it only considers mimics from large dsDNA viruses, and largely from herpesviruses. The practical reasoning behind this is obvious, but it is important to point out that mimics from RNA viruses, which evolve much faster than DNA viruses, and also cross species much more frequently than herpesviruses, may apply different selection pressure on host genomes.

Other comments:

1) The sentence on page 14 that begins "When comparing the fraction of essential genes ..." is not a complete sentence.

Reviewer #2:

This paper presents an interesting and valuable study of the roles of positive selection and evolutionary constraint related to interactions between host proteins and viral proteins that mimic host protein interaction partners, integrating evolutionary and computational molecular analysis. This is presented as a question of whether there is an evolutionary arms race between the host proteins and viruses, ultimately finding that host proteins targeted by viral mimics are under more constraint than non-target proteins, and that the interaction sites specifically are under more constraint than other protein regions. In particular, I find the result that the evolutionary rate of host proteins is lower in the interaction regions with viral mimics and other host proteins than the evolutionary rate the non-interaction regions to be compelling evidence for this point. I believe this is a valuable contribution to the literature and of interest to a broad audience including evolutionary biologists, functional genomicists, molecular biologists, and immunologists. However there are two interrelated points that I feel should be addressed:

- The paper does not discuss the functions of host proteins, beyond noting that some but not most of the proteins are involved in immune processes, and that "similarities to ... immunoglobulin domains results in many homologs" (p. 5). While not the focus of the question of an evolutionary arms race, the functions of the involved proteins may be important to take into account, including in situations in which proteins of certain types resulting in more structural homolog pairs than others, since this creates a potential source of bias in the analysis.

- Related to this, it is not clear whether it is the types of proteins targeted by mimics that are conserved in general, or that mimics target conserved proteins specifically. In other words, do viral mimics target proteins with specific functions (that happen to be conserved), or are viral mimics only successful when they target conserved proteins (regardless of protein function)?

The overall impact of these two issues is that the results may be biased by the functions of host proteins involved in these interactions. It should be practical to control for this in the analyses presented, either by bootstrapping sets of control genes with shared GO processes (as described in Enard et al. 2016) or more generally by grouping host proteins by major functions and comparing only within groups (here referring to analyses presented in fig. 2). A related issue is brought up in the discussion about viral mimics (p. 20), that "we do not account for many other functions that can contribute to the evolutionary rates of viral proteins. Furthermore, other factors not directly related to mimicry... may influence evolutionary rates." These are important points that are not sufficiently addressed. For example, how likely is it that these additional functions and factors would change the results of the analyses? The functional questions could be addressed in part by comparing the overall constraint among (viral or host) genes with different (combinations of) GO processes to assess whether the presented analyses are conservative and thus likely robust to this issue, or not. Fold complexity, core vs accessory functions, and possibly time of expression (all possible factors brought up in this section) could likely be similarly assessed.

The second point, whether viral mimics are targeting conserved regions and proteins specifically, or whether they are targeting proteins of certain functions that happen to be conserved (what are host targets of viral mimics, typically?), has additional implications for interpreting the co-evolutionary relationship between host and virus, and thus the presence or absence of evidence of an evolutionary arms race.

Minor comments

- In general, while it is obvious in the figures, it is not always clear right away what tests rise to the level of statistical significance in the text. For example, in the section "Few host targets of viral mimics have evidence of pervasive positive selection," the first half of the first paragraph details the differences between tested groups in their percent of proteins with positive selection, and only mentions toward the end that "none of the groups is significantly enriched with, or depleted of, positively significant proteins..." (p8). This makes it unnecessarily difficult to interpret the true results of these tests.

- P8, "This is likely due to the small numbers of positively selected proteins in each group, and the overall small number of proteins in some of the tested groups." It would be helpful to include, either in a figure or in a supplementary table, the number of proteins analyzed in these various groups, to contextualize the lack of statistically significant enrichment.

- Another example of limited or obscured information about statistical significance is p11, "In mimicked proteins, the general interface has a slightly higher fraction of PSRs than the mimicry interface..."

- The number of genes tested is listed differently in different parts of the text. On p25, "a total of 10,544 genes" before filtering, and on p26, "After filtering, 9,178 genes." However for the "Evolutionary Rate Inference" section, 10,531 genes were used (10,544 minus 13 genes that failed) - why were unfiltered genes used? Furthermore, in the "Positive selection analysis" section, 10,540 genes are mentioned in the first sentence, and later in the paragraph, "After filtering, 8,664 genes remained." It is also unclear in the results whether the filtered or unfiltered MSAs were used (p7, "removing MSAs with too many masked regions, resulting in a set of 10,544 MSAs.")

Reviewer #3:

This is an interesting manuscript on the computational analysis of the evolutionary interplay between host and viral binding interfaces, with the interesting conclusion that host protein interfaces that are mimicked by viruses evolve slowly, and that viral proteins that mimic the host proteins are not rapidly evolving. This contrasts with the general idea of the arms race between host and viral proteins. The study seems to be well performed, and the manuscript is easy to read and is easy to follow,

although it is a bit wordy.

Not being a computational expert, I can not comment on the analysis performed, so this needs to be assessed by a reviewer with competence in the area.

Point-by-point response to reviewers' comments -The evolutionary dynamics between viral mimics and host proteins

We would like to thank the editors and the reviewers for their helpful and constructive comments and for taking the time to evaluate our manuscript. In the revised manuscript we have thoroughly addressed the reviewers' questions, as detailed below (in blue). These modifications lead to the addition of two panels in Main Fig 2 (regarding functional groups and their conservations in host mimicked and target proteins), to the addition of 2 supporting figures (related to various characteristics of viral mimicking vs non-mimicking proteins), and to 3 additional tables (of viral proteins UNIPROT IDs, GO terms analysis, and mimicry groups proteins).

Finally, during this revision, we noticed that there were a few proteins with dN/dS values that were wrongly excluded from the original analyses in Fig 3A & 3C (2 mimicked and 3 targets), and that in Fig 2A and 4D we included a group of host proteins with high over-masked residue fractions – all of these were now corrected in the relevant analyses and in the modified figures (the corrected numbers of proteins / residues appear in cyan next to the original numbers that appear with a strikethrough in the Fig Legends and in the Results section). All the conclusions and trends reported in the original manuscript remain the same, including the statistical significance.

Reviewer #1:

Fuchs et al. perform evolutionary analyses on a large number of viral mimics and the host proteins they interact with and mimic structurally. Starting with the proteomes of five large dsDNA viruses, they use structural modeling to predict a set of viral proteins that mimic host proteins, and go on to predict the host proteins that these viral mimics likely interact with. This defines a set of host genes involved in viral mimicry that they perform evolutionary analyses on to determine how these evolve across primates compared to other genes that are either known to interact with viruses or not. They further define a set of 59 predicted host-virus complexes that they model structurally to analyze the evolution of residues at, or away from, the predicted virus-host interaction interfaces. Finally, they show that viral mimics are relatively well conserved within viral genomes, especially in the interface residues that interact with host proteins.

Overall, this is a well performed set of analyses and represents a large dataset of evolutionary data on viral mimics and the host proteins they mimic or interact with. While several of the conclusions have been proposed previously for individual virus mimic examples, the size of the dataset lends credence to the generalizable nature of the conclusions. Importantly, the authors also make several efforts to caveat their conclusions so as to not over-interpret their data,

although one important caveat that is not raised is the reliance on DNA viruses, and mostly herpesviruses, exclusively. Additional comments to improve the readability or utility of the paper are described below.

Major concerns:

1) Several points in the manuscript could benefit from being more explicit about how analyses were performed, either in the results of the methods sections. For instance, on page 5, the authors describe 540 viral proteins that went into their analysis. Where did this list come from? Can the authors include this list in a supplemental table? Similarly, on page 10, they describe using AlphaFold-multimer and getting a list of 59 protein complexes, but it is unclear how many were used as starting points. In general, it would help readability to be as transparent as possible with what the starting and ending lists are for all of these analyses.

Following the reviewer's suggestions, we now added a new supplemental table (Table EV1) that details the list of all 540 viral proteins with their UNIPROT accession and viral species.

We also added another supplemental table (Table EV3) that includes the list of all human proteins and for each protein whether it is defined as a structural mimicked, functional mimicked, viral binding and/or target protein in this study.

Furthermore, we added in Results additional details on numbers of proteins / complexes / residues we mention in different sections of the manuscript:

a) On page 5 we added a clarification on the group of 540 proteins -

"To find host-like domains in viral proteomes, we used AlphaFold2(Tunyasuvunakool *et al*, 2021), and predicted the structures of 540 viral proteins (from the proteomes of the five viruses used in the analysis, downloaded from Uniprot – see additional details in Methods). The list of all viral proteins and their accession IDs appears as Table EV1. "

b) On page 5, we added the numbers of all the human-viral protein structural matches before any filtering stage, 33,011 matches (this is the starting set that after filtering resulted in 7,593 matches, mentioned in the manuscript several lines afterwards):

"We then filtered matching 33,011 human-viral structural homologs, using a series of filtering stages..."

c) On page 6, we added the following to clarify the flow of analysis and what was the starting point of finding mutual targets:

"Overall, we identified 645 mimicked-mimicking-target triads, out of 7,593 homologous pairs."

d) Regarding the 59 protein complexes the reviewer mentioned - on page 11, we specified the starting point for AlphaFold-Multimer runs (for both pairs of mimicked-targets and mimicking-targets):

"To identify interface regions between host-mimicked and target proteins, we employed AlphaFold-multimer(Evans *et al*, 2021) with 576 mimicked-target pairs and used the resulting 59 protein complexes whose prediction scores were sufficiently high ($0.8 \times ipTM + 0.2 \times pTM > 0.7$). We also used 12 complexes of viral-mimicking and host-target proteins with high-enough prediction scores (out of 116 pairs), to similarly identify interface regions between viral mimics and host target proteins."

e) In the legend of Fig 3 (page 40) that is relevant to the protein complex analysis, we replaced the text to have a more detailed version of how many proteins and residues were used (based on whether they had evolutionary rates).

We also note that Table EV4 (originally Table S2) includes details on various groups of proteins and complexes used in the manuscript, along with the relevant Figure number. This table was expanded in the revision, and now also includes numbers relevant to the analysis of the 59 complexes.

2) The authors describe their Alphafold confidence threshold as $(ipTM + pTM > 0.7)$. Such a threshold would be very low, as each of those values could be < 0.4 . If that is the case, the threshold needs to be higher. However, I am assuming the authors mean that $ipTM < 0.7$ and $pTM < 0.7$. If so, please clarify.

The confidence threshold was indeed not adequately described. The actual formula used was $(0.8 * ipTM + 0.2 * pTM) > 0.7$. This calculation is taken directly from the AlphaFold output and this score is generally considered an indication for a high-quality prediction, e.g., Shor & Schneidman-Duhovny, *Nat Methods*, 2024. We now corrected this in the manuscript (Pages 11 and 29).

3) The ScanNet program, and exactly what it is predicting, needs to be described in more detail given how it is being used to "predicts binding sites on protein surfaces". Is there an alternate

way to predict binding sites on proteins? I appreciate that the authors set several different thresholds for their ScanNet outputs, but it appears to be largely a black box in terms of what is exactly being predicted, which is insufficient to then draw conclusions from.

We now also used ISPRED4 (Savojardo et al, Bioinformatics, 2017) as an alternative interface predictor that uses a different approach than ScanNet. The results obtained with the new method are in strong agreement with the previous observations, based on ScanNet (see Fig EV2, new panel D).

We added additional details to describe both methods (ScanNet and ISPRED4) and their input to the relevant sections in Results (Page 11):

"This was done using two methods for predicting binding sites on protein surfaces, based on the protein monomeric structure: ScanNet(Tubiana *et al*, 2022), a recently developed program that predicts binding sites on protein surfaces, based on a geometric deep learning model that uses the spatio-chemical arrangement of amino acids and their neighbors, and ISPRED4(Savojardo et al. 2017), that combines Support Vector Machines with Conditional Random Fields (CRF) to predict interface sites."

And in Methods (Page 30):

"In order to identify interface regions of host-mimicked proteins or of target proteins with other proteins, unrelated to mimicry, we used the ScanNet program(Tubiana *et al*, 2022). ScanNet is a geometric deep learning model for prediction of protein-protein interaction sites. We used the following parameters to define residues as belonging to interface regions – a minimal score of 0.7 across all residues for Main Fig 3C, and a minimal score of 0.7 or 0.5, with all residues or only with ordered residues, for Supp Fig S3 EV2. These interface regions are defined as “general-interface” regions in the relevant sections. We further used the ISPRED4 program(Savojardo et al. 2017) as an additional method to infer interface regions. This method uses a different model than ScanNet, based on the combination of support vector machines (SVMs) and grammar-restrained conditional random fields (GRHCRF). The program was run using default parameters, and a minimal interface probability of 0.7 was required for a residue to be defined as interface."

4) As mentioned in the summary, the one major caveat to this study is that it only considers mimics from large dsDNA viruses, and largely from herpesviruses. The practical reasoning behind this is obvious, but it is important to point out that mimics from RNA viruses, which evolve much faster than DNA viruses, and also cross species much more frequently than herpesviruses, may apply different selection pressure on host genomes.

We agree that this is a very important point. We now added the following to Discussion, in the section on the limitations of our analysis (Page 21):

"Mimics encoded by RNA viruses, that evolve much faster than DNA viruses and can cross between host species more frequently than herpesviruses, may have different evolutionary dynamics with their host. "

Other comments:

1) The sentence on page 14 that begins "When comparing the fraction of essential genes ..." is not a complete sentence.

We now corrected this sentence – thank you for noticing!

"~~When comparing the fraction of essential genes across these groups, we~~ We also observe that the group of viral-binding human proteins has the highest fraction of essential genes among the tested groups"

Reviewer #2:

This paper presents an interesting and valuable study of the roles of positive selection and evolutionary constraint related to interactions between host proteins and viral proteins that mimic host protein interaction partners, integrating evolutionary and computational molecular analysis. This is presented as a question of whether there is an evolutionary arms race between the host proteins and viruses, ultimately finding that host proteins targeted by viral mimics are under more constraint than non-target proteins, and that the interaction sites specifically are under more constraint than other protein regions. In particular, I find the result that the evolutionary rate of host proteins is lower in the interaction regions with viral mimics and other host proteins than the evolutionary rate the non-interaction regions to be compelling evidence for this point. I believe this is a valuable contribution to the literature and of interest to a broad audience including evolutionary biologists, functional genomicists, molecular biologists, and immunologists. However there are two interrelated points that I feel should be addressed:

- The paper does not discuss the functions of host proteins, beyond noting that some but not most of the proteins are involved in immune processes, and that "similarities to ... immunoglobulin domains results in many homologs" (p. 5). While not the focus of the question of an evolutionary arms race, the functions of the involved proteins may be important to take into account, including in situations in which proteins of certain types resulting in more structural homolog pairs than others, since this creates a potential source of bias in the analysis.
- Related to this, it is not clear whether it is the types of proteins targeted by mimics that are conserved in general, or that mimics target conserved proteins specifically. In other words, do viral mimics target proteins with specific functions (that happen to be conserved), or are viral mimics only successful when they target conserved proteins (regardless of protein function)?

The overall impact of these two issues is that the results may be biased by the functions of host proteins involved in these interactions. It should be practical to control for this in the analyses presented, either by bootstrapping sets of control genes with shared GO processes (as described in Enard et al. 2016) or more generally by grouping host proteins by major functions and comparing only within groups (here referring to analyses presented in fig. 2). A related issue is brought up in the discussion about viral mimics (p. 20), that "we do not account for many other functions that can contribute to the evolutionary rates of viral proteins. Furthermore, other factors not directly related to mimicry... may influence evolutionary rates." These are important points that are not sufficiently addressed. For example, how likely is it that these additional functions and factors would change the results of the analyses? The functional questions could be addressed in part by comparing the overall constraint among (viral or host) genes with different (combinations of) GO processes to assess whether the presented analyses are conservative and thus likely robust to this issue, or not. Fold complexity, core vs accessory functions, and possibly time of expression (all possible factors brought up in this section) could likely be similarly assessed.

With respect to host proteins mimicked or targeted by viral mimicking proteins:

We now added a functional enrichment analysis for these two categories ("mimicked" and "target" proteins). This provided a nice addition to the manuscript, revealing that both host-mimicked and target protein classes tend to be involved in various antiviral pathways (such as "cell death", "response to cytokine", "inflammatory response") and in pathways often modulated by viruses ("regulation of Golgi", "G1/S transition", "signal transduction"). In each GO term category, we tested whether proteins related to mimicry are more conserved than the rest of the human proteins belonging to this GO term. We observed that in functional

mimicked, in 158 out of 209 enriched terms (75.6%), the mimicked are significantly more conserved than non-mimicked belonging to the same category. In host targets, in 36 out of 57 enriched terms (63.2%) the targets are significantly more conserved than non-targets. This suggests that in the majority, but not in all cases, host proteins related to mimicry are more conserved than other proteins with similar functions.

We have added this analysis and its details to Results (Page 10) and Methods (Page 29). The conservation results of the lists of non-redundant GO terms appear as panels F and G in Figure 2, for targets and mimicked proteins, respectively. The complete lists of all enriched terms (including redundant terms) and their relative conservations with FDR-corrected P-values appear as Table EV6.

The added text in Results is:

"Next, we used GO term analysis to test which functions are enriched in host proteins related to mimicry. Both functional mimicked and targets are enriched with various antiviral pathways ("cell death", "response to cytokine", "inflammatory response") and in pathways often modulated by viruses ("regulation of Golgi", "G1/S transition of mitotic cell cycle", "signal transduction"). In each GO term category, we tested whether proteins related to mimicry are more conserved than unrelated proteins (**Fig 2F-G and Table EV6**). We observed that functional mimicked are more conserved than non-mimicked proteins belonging to the same GO category in 158 out of 209 (75.6%) enriched terms. Targets are more conserved than non-targets in 36 out of 57 (63.2%) enriched terms. Thus, in the majority but not in all cases, mimicked and target proteins are more conserved than respective host proteins within the same functional category."

The added text to Methods is:

Functional enrichment analysis

We used gProfiler(Raudvere et al. 2019) to test for functional enrichment of host targets and mimicked proteins against all human proteins. For each of the significantly enriched terms in the biological processes category, we then tested whether targets or mimicked proteins are more conserved than the rest of the human proteins belonging to this term, using Mann-Whitney test and correcting using FDR (**Table EV6**). From these lists, non-redundant term lists

are shown with their dN/dS distributions in Figure 2F-G (taking the most significant term from each group of related GO terms)."

With respect to viral mimicking proteins and their functions:

Functional enrichment analysis with GO terms, as done in the case of host proteins above or as described in Enard et al. 2016, is unlikely to yield significant or meaningful results when performed with viral mimicking proteins. This is because the overall protein numbers in these viral genomes is significantly smaller than in the host, and the numbers of mimicking proteins with evolutionary data is even smaller (the total number of mimicking proteins with evolutionary data ranges between 6 in HCMV to 36 in VACV). Furthermore, GO terms are partial and inaccurate for many viral proteins (because of fewer functional studies and because the annotations are not as comprehensive as in human genes). Thus, there would be very few viral proteins per category and the set of proteins would not be necessarily accurate, not allowing for a statistically rigorous examination.

Instead, we added an analysis of four different factors thought to be related to viral protein evolutionary rates including (1) core versus accessory proteins and gene evolutionary age (as defined in two previous studies), (2) fold complexity, (3) temporal gene expression, and (4) protein disorder content (we recently shown that the latter three characteristics are associated with rates of herpesvirus protein evolution – Fuchs, Mozzi, et al, *MBE* 2025).

When comparing mimicking versus non-mimicking proteins using each of the above-mentioned characteristics, we did not observe consistent trends across viruses (e.g., that mimicking proteins tend to have higher fold complexity than non-mimicking across all tested viruses), and only a few of the comparisons were statistically significant. We added these analyses to Results and as two supplemental figures, and mentioned that we cannot conclude that any of these factors bias the results based on these analyses (Page 16-17):

"We tested whether mimics differ from non-mimics in various structural and functional characteristics thought to be related to evolutionary rates of these viral proteins(Fuchs et al. 2025; Molteni et al. 2023) (Fig EV5 and Appendix Fig S2). However, none of these comparisons supported the notion that the evolutionary rates of mimics are biased by these characteristics in a consistent manner across all studied viruses."

The technical details of these analyses were added to a new section in Methods (Pages 31-32):

"Analysis of viral gene and protein characteristics".

The results themselves appear as Fig EV5 (core vs accessory, gene age, temporal gene expression), and Appendix Fig S2 (fold complexity and fraction of disordered regions).

Minor comments

- In general, while it is obvious in the figures, it is not always clear right away what tests rise to the level of statistical significance in the text. For example, in the section "Few host targets of viral mimics have evidence of pervasive positive selection," the first half of the first paragraph details the differences between tested groups in their percent of proteins with positive selection, and only mentions toward the end that "none of the groups is significantly enriched with, or depleted of, positively significant proteins..." (p8). This makes it unnecessarily difficult to interpret the true results of these tests.

Following the reviewer comment, we moved the statements of the significance to the beginning of this section and shortened it (Page 8):

"While the groups tested differ in their fraction of PSGs, none is significantly depleted or enriched with respect to the set of all human proteins, likely due to the small numbers of PSGs in each group (ranging from 0 to 494, in target and all proteins, respectively)."

- P8, "This is likely due to the small numbers of positively selected proteins in each group, and the overall small number of proteins in some of the tested groups." It would be helpful to include, either in a figure or in a supplementary table, the number of proteins analyzed in these various groups, to contextualize the lack of statistically significant enrichment.

We now added the following sentence, including all group sizes in different FDR cutoffs, to the legend of the relevant figure (Fig EV1):

"The total numbers of PSGs in each of the categories are for all human, viral binding, target and functional mimicked, respectively, in: FDR-corrected P-values of 0.0001: 211,103,0,3; FDR-corrected P-values of 0.001: 301,137,0,5; and FDR-corrected P-values of 0.05: 766,338,1,10."

The PSG numbers in FDR corrected P-values of 0.01 appear in Fig2 legend, since they are related to this panel.

- Another example of limited or obscured information about statistical significance is p11, "In mimicked proteins, the general interface has a slightly higher fraction of PSRs than the mimicry interface..."

We now added the total numbers of PSRs and all residues to this section, for each of three instances (Page 12-13).

- The number of genes tested is listed differently in different parts of the text. On p25, "a total of 10,544 genes" before filtering, and on p26, "After filtering, 9,178 genes." However for the "Evolutionary Rate Inference" section, 10,531 genes were used (10,544 minus 13 genes that failed) - why were unfiltered genes used? Furthermore, in the "Positive selection analysis" section, 10,540 genes are mentioned in the first sentence, and later in the paragraph, "After filtering, 8,664 genes remained." It is also unclear in the results whether the filtered or unfiltered MSAs were used (p7, "removing MSAs with too many masked regions, resulting in a set of 10,544 MSAs.")

The original description of the pipeline was indeed unclear and confusing in terms of when and which filtering was performed and what set was used. We only used the set of filtered genes in each of the analyses shown in the manuscript (however, for technical reasons, we ran all genes in PAML and Selecton, and later removed those that were filtered due to over-masking from subsequent analyses). We now clarified it as follows in Methods (Pages 25-27):

We moved the description of the MSA filtering ("Next, we filtered out genes...") from the end of the section "**Sequence alignment and tree reconstruction**" to the end of the section "**Evolutionary rate inference**", which is the stage in the analysis that this filtering took place.

We also modified the last sentence in this section to: "After filtering, 9,178 genes remained, and were included in the analysis."

Thus, it should now be clear that the rate inference analysis was performed on 9,178 genes, after filtering over-masked genes.

In the subsequent section "**Positive selection analysis**", we also used the set of genes after filtering. However, here a stricter filtering was used, removing genes with high fractions of either masked or gapped residues (as mentioned in the text). Thus, only 8,664 genes were included in this specific analysis.

We rephrased several sentences, to make it clearer (Pages 26-27):

In the beginning of this section:

"We analyzed a set of 10,540 human genes. Of the 10,544 human genes with MSAs, 10,540 were analyzed by PAML (4 genes have failed during the run). Next, we filtered out genes whose MSAs were over-masked."

In the end of this section:

"After filtering, 8,664 genes were included in the analysis."

It should now be clear that we started with a larger set of genes and then used a smaller set for the analysis, following filtering.

Reviewer #3:

This is an interesting manuscript on the computational analysis of the evolutionary interplay between host and viral binding interfaces, with the interesting conclusion that host protein interfaces that are mimicked by viruses evolve slowly, and that viral proteins that mimic the host proteins are not rapidly evolving. This contrasts with the general idea of the arms race between host and viral proteins. The study seems to be well performed, and the manuscript is easy to read and is easy to follow, although it is a bit wordy.

Not being a computational expert, I can not comment on the analysis performed, so this needs to be assessed by a reviewer with competence in the area.

Following the reviewer's comments, we revised the Discussion to shorten it – sentences that were shortened or removed appear with a strikethrough in the revised Discussion (Pages 18-21).

30th Jan 2026

Manuscript Number: MSB-2025-13373R

Title: The evolutionary dynamics between viral mimics and host proteins

Author: Rotem Fuchs

Ofir Schor

Bar Naim

Dafna Tussia-Cohen

Alessandra Mozzi

Diego Forni

Sivan Friedman

Zohar Haggai

Manuela Sironi

Tzachi Hagai

Dear Prof Hagai,

Thank you for sending us your revised manuscript. We have now heard back from the two reviewers who were asked to re-evaluate your study. As you will see, the reviewers are satisfied with the modifications made. Before we can formally accept your manuscript, we would ask you to address the following editorial-level issues:

1. Please add a missing callout for Fig 1F.
2. Include a "Disclosure and competing interests statement".
3. Please remove the synopsis text (standfirst and bullet points) from the manuscript file and upload it in a separate Word document. Also, remove the running title from the manuscript.
4. Appendix: Please add page numbers for all items in the Table of Contents and ensure they appear on each page throughout the document.
5. "Materials and Methods" should be renamed to "Methods". "Data and code availability" should be renamed to "Data availability".
6. Currently, there are 7 EV tables provided in a single zip folder. Each file needs to be uploaded separately. Table EV4 and Table EV7 are the only tables that will remain as EV tables, but they must be renamed to Table EV1 and Table EV2 and uploaded as Expanded View Content. The other tables are more complex and should be renamed as datasets, updated to Dataset EV1-Dataset EV5, and uploaded accordingly.

For all tables and datasets:

- Remove the legends from the manuscript file.
- Include the legend in the corresponding Excel file. For EV tables, place the legend in the same sheet. For datasets, provide the legend in a separate sheet/tab.

All renaming must be applied consistently across source file names, legends, titles in the system, callouts in the manuscript.

7. Source Data: Please note that source data need to be uploaded as one zipped file for each figure.
8. Please address the following issues related to figure legends:
 - Please note that the legend for figure EV2 D is missing in the manuscript. This needs to be rectified.
 - Please define the annotated p values ****/****/**/ as well as provide the exact p-values for the same in the legend of figure 3A, C, D; 4B, D; 5E, F as appropriate.
 - Please note that the exact p values are not provided in the legends of figures 1A, C, D; EV2 A-D; EV3 B, C; EV4 A-D
 - Please note that the box plots need to be defined in terms of minima, maxima, centre, bounds of box and whiskers, and percentile in the legends of figures 1A, D; 3A, C; 4B, D; 5E, F; EV2 A-D; EV 3 C, EV4 A-D
 - Please note that information related to n is missing in the legends of figures 3A, C; 5E, F; EV2 A-D; EV4 A-D
 - Please note that the error bars are not defined in the legends of figures 1C, EV3 B

Kind regards,
Jingyi

Jingyi Hou, PhD
Senior Editor
Molecular Systems Biology

*** PLEASE NOTE *** As part of the EMBO Press transparent editorial process initiative (see our Editorial at <https://dx.doi.org/10.1038/msb.2010.72> , Molecular Systems Biology will publish online a Review Process File to accompany accepted manuscripts. When preparing your letter of response, please be aware that in the event of acceptance, your cover letter/point-by-point document will be included as part of this File, which will be available to the scientific community. More information about this initiative is available in our Instructions to Authors. If you have any questions about this initiative, please contact the editorial office (msb@embo.org).

Reviewer #1:

The revised manuscript addresses all of my previous concerns.

Reviewer #2:

The authors addressed my comments thoroughly and well for this revision. I have nothing further.

Point-by-point response to editor's request

We would like to thank the editors and the reviewers for their helpful comments. We corrected and modified the manuscript files as detailed below, with respect to the Editor's requests.

1. Please add a missing callout for Fig 1F.

We have added it to Main text, page 7:

We next employed phylogeny-based approaches to study coding-sequence evolution of all human protein-coding genes across primates, to compare evolutionary trends between genes encoding for host-mimicked and mutual target proteins, and other human protein groups (**Fig 1F**).

2. Include a "Disclosure and competing interests statement".

This was added to the end of Page 31.

3. Please remove the synopsis text (standfirst and bullet points) from the manuscript file and upload it in a separate Word document. Also, remove the running title from the manuscript.

This was removed.

4. Appendix: Please add page numbers for all items in the Table of Contents and ensure they appear on each page throughout the document.

We added page numbers to the revised Appendix.

5. "Materials and Methods" should be renamed to "Methods". "Data and code availability" should be renamed to "Data availability".

These sections were renamed.

6. Currently, there are 7 EV tables provided in a single zip folder. Each file needs to be uploaded separately. Table EV4 and Table EV7 are the only tables that will remain as EV

tables, but they must be renamed to Table EV1 and Table EV2 and uploaded as Expanded View Content. The other tables are more complex and should be renamed as datasets, updated to Dataset EV1-Dataset EV5, and uploaded accordingly.

We have renamed the original 7 EV Tables as requested.

For all tables and datasets:

- Remove the legends from the manuscript file.

The legends were removed from the main doc file.

- Include the legend in the corresponding Excel file. For EV tables, place the legend in the same sheet. For datasets, provide the legend in a separate sheet/tab.

All renaming must be applied consistently across source file names, legends, titles in the system, callouts in the manuscript.

We have added the legends as requested in the Excel sheets. The text was renamed accordingly.

7. Source Data: Please note that source data need to be uploaded as one zipped file for each figure.

We now uploaded them as separate zip files (named - Figure 1,2,3,4,5).

8. Please address the following issues related to figure legends:

- Please note that the legend for figure EV2 D is missing in the manuscript. This needs to be rectified.

We now added:

"(D) ISPREd4 probability of 0.7 and above is used to determine interface residues (all residues were considered)."

- Please define the annotated p values ****/**/*/* as well as provide the exact p-values for the same in the legend of figure 3A, C, D; 4B, D; 5E, F as appropriate.

We added the following definition to these legends:

"In all panels: ***P < 0.001, **P < 0.01, *P < 0.05."

- Please note that the exact p values are not provided in the legends of figures 1A, C, D; EV2 A-D; EV3 B, C; EV4 A-D

We assume the request regarding "Figure 1A,C,D" referred to Figure 2A,C,D.

We added exact P-values to all mentioned panels.

- Please note that the box plots need to be defined in terms of minima, maxima, centre, bounds of box and whiskers, and percentile in the legends of figures 1A, D; 3A, C; 4B, D; 5E, F; EV2 A-D; EV 3 C, EV4 A-D

We added the following definition to these legends:

Boxplots represent the median, first quartile and third quartile with lines extending to the furthest value within 1.5 of the interquartile range (IQR).

We assume the request regarding "Figure 1A, D" referred to Figure 2A, D.

- Please note that information related to n is missing in the legends of figures 3A, C; 5E, F; EV2 A-D; EV4 A-D

We added exact N size to all mentioned panels.

- Please note that the error bars are not defined in the legends of figures 1C, EV3 B

The mentioned figure panels are violin plots – we have added the following description in 2C, EV3B - each filled area extends to represent the entire data range. "

10th Feb 2026

Manuscript number: MSB-2025-13373RR

Title: The evolutionary dynamics between viral mimics and host proteins

Dear Prof Hagai,

Thank you again for sending us your revised manuscript. We are now satisfied with the modifications made and I am pleased to inform you that your paper has been accepted for publication.

You may qualify for financial assistance for your publication charges - either via a Springer Nature fully open access agreement or an EMBO initiative. Check your eligibility: <https://link.springer.com/journal/44320/how-to-publish-with-us>

Sincerely,
Jingyi

Jingyi Hou, PhD
Senior Editor
Molecular Systems Biology

>>> Please note that it is Molecular Systems Biology policy for the transcript of the editorial process (containing referee reports and your response letter) to be published as an online supplement to each paper. If you do NOT want this, you will need to inform the Editorial Office via email immediately. More information is available here: <https://link.springer.com/partners/embo-press/editorial-policies#Peer%20review>